# Locating What You Need: Towards Adapting Diffusion Models to OOD Concepts In-the-Wild

Jianan Yang[1,2]     Chenchao Gao[4,2]     Zhiqing Xiao[1,2]     Junbo Zhao[1,2]     Sai Wu[1,2]
Gang Chen[1,2], Haobo Wang[3,2*]

[1]College of Computer Science and Technology, Zhejiang University
[2]Hangzhou High-Tech Zone (Binjiang) Institute of Blockchain and Data Security
[3]School of Software Technology, Zhejiang University
[4]International School of Information Science and Engineering, Dalian University of Technology
{jianan0115,zhiqing.xiao,j.zhao,wusai,cg,wanghaobo}@zju.edu.cn
gccgyllyp@gmail.com

## Abstract

The recent large-scale text-to-image generative models have attained unprecedented performance, while people established *adaptor* modules like LoRA and DreamBooth to extend this performance to even more unseen concept tokens. However, we empirically find that this workflow often fails to accurately depict the *out-of-distribution* concepts. This failure is highly related to the low quality of training data. To resolve this, we present a framework called **C**ontrollable **A**daptor **T**owards **O**ut-of-**D**istribution Concepts (CATOD). Our framework follows the active learning paradigm which includes high-quality data accumulation and adaptor training, enabling a finer-grained enhancement of generative results. The *aesthetics* score and *concept-matching* score are two major factors that impact the quality of synthetic results. One key component of CATOD is the weighted scoring system that automatically balances between these two scores and we also offer comprehensive theoretical analysis for this point. Then, it determines how to select data and schedule the adaptor training based on this scoring system. The extensive results show that CATOD significantly outperforms the prior approaches with an 11.10 boost on the CLIP score and a 33.08% decrease on the CMMD metric.

## 1 Introduction

The generative modeling for text-to-image has attained unprecedented performance most recently [37, 45, 41, 49]. Notably, by training over billions of text-image data pairs [52, 51], the family of diffusion models has allowed high-fidelity image synthesis directed by the *prompt* provided in production. Despite their massive successes, these models still fail to generate images with decent quality and matching semantics, when encountering prompts that contain unseen or *out-of-distribution* concept tokens [60, 12, 21]. Simply put, the reason causing this failure is due to that the training set is **not** unbounded with limited variations. On the side of production, this limitation could significantly impact the practicality of this technique in real-world applications.

To deal with such concepts, recent works have resorted to *adaptors* such as Textual Inversion [18], DreamBooth [47, 58, 48], and LoRA [25, 66, 76], which tunes only a small part of the text-to-image model or insert extra modules. These adaptors largely reduce the training costs, and more importantly, preserve the visual aesthetic information originally learned by the underlying model.

---

[*]Corresponding author.

38th Conference on Neural Information Processing Systems (NeurIPS 2024).

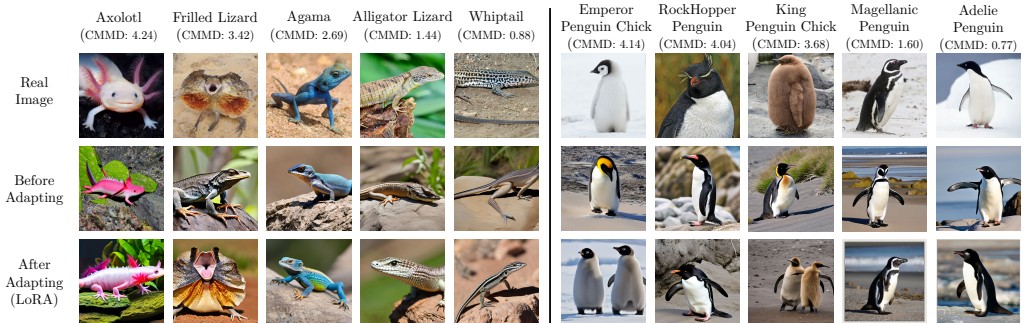

Figure 1: **Comparison of images generated before/after training adaptors over concepts with different CMMD scores.** One observation is that concepts with higher CMMD scores are notably more challenging for the underlying model to generate (the second row). Additionally, we also notice that a higher CMMD value leads to a more notable loss of visual details when training adaptors (the third row).

However, in this paper, we have found that recent works still struggle to accurately depict the visual details of *out-of-distribution* concepts (with a CMMD score above 3.5), as illustrated in Figure 1. Adapters like LoRA are able to accurately represent the shape and color of OOD concepts compared to the generative results before adaptor training, but they fall short when it comes to finer details such as texture, contours, and patterns. This issue arises from the fact that current studies predominantly focus on variations of in-distribution (ID) concepts (e.g., humans, dogs, and cats) while ignoring the out-of-distribution (OOD) ones. This failure motivates us to think about what makes the problem of distorted visual details happen when training adaptors.

In Figure 2, we observed that how an adaptor depicts OOD concepts can be significantly influenced by the quality of the training data. If the model is trained on samples containing disruptive objects, the resulting generative outputs are likely to reflect these disruptive elements. When the training data contains images with vague or very small instances of the OOD concept, the generative results may appear low-quality. In contrast, the high-quality data for adapting OOD concepts usually contains a single and clear object corresponding to the given concept, which is highly distinguishable from the background and other types of objects and helps produce accurate results with high-fidelity. However, manually picking such high-quality data requires much human labor and expertise, which may crucially limit the versatility of text-to-image models. Therefore, an effective paradigm to locate high-quality samples of OOD concepts is important for the practical shipping of this field.

To this end, we developed a framework called **C**ontrollable **A**daptor **T**owards **O**ut-of-**D**istribution Concepts (referred to as CATOD) which aims to identify high-quality samples to guide the adaptor training. This framework follows the Active Learning (AL) paradigm [54, 44], involving iteratively accumulating training data and updating the adaptor. The profound motivation of this approach is to comprehensively model the interaction between training data and the underlying text-to-image model. Specifically, CATOD includes two interconnected scores: the *aesthetic* score and the *concept-matching* score, following the observation that object clarity and uniqueness largely impact generative results, as illustrated in Figure 2. Based on this, we devised a weighted scoring system that adapts itself according to the adaptor to select high-quality data while also properly balancing the two scores. With the carefully selected high-quality data, we schedule the adaptor training based on the quality of generative results evaluated through this scoring system.

In summary, our contributions are as follows: (i)-We have identified the challenge of adapting text-to-image models to out-of-distribution (OOD) concepts, where recent studies often struggle to accurately depict them; (ii)-We have introduced a framework called CATOD that iteratively updates training data and the adaptor to generate OOD concepts precisely; (iii)-Our extensive experiments verified that CATOD achieves significant performance gain with up to 11.10 on the CLIP score and 33.08% on the CMMD metric; (iv)-We have also offered theoretical insights into the key factors: *aesthetics* and *concept-matching*, which contribute to the effectiveness of our method.

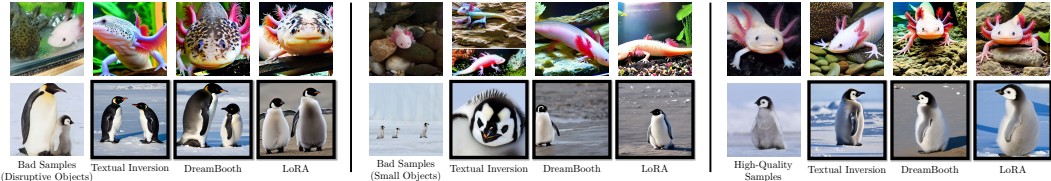

Figure 2: **Comparison of synthetic results on data with different quality.** Generated images are significantly influenced by the quality of training data. If the training data includes disruptive objects, the generative images may include disruptive visual details (*Left*). When an object within the image is too small, the results may not accurately represent the intended concepts (*Middle*). In contrast, if the image contains a high-fidelity object without disruptive elements (*Right*), the model is more likely to generate the desired result accurately.

## 2 OOD Problem in Latent Diffusion Models

**Revisiting Latent Diffusion Models (LDMs).** Latent Diffusion Models (LDMs) [45] comprise two components: a diffusion process operating the latent space and an auto-encoder which contains an encoder $\mathcal{E}$ mapping an image into the latent space and a decoder $\mathcal{D}$ that reconstruct images from latent codes. Furthermore, the diffusion process can be conditioned on the output of text embedding models, enabling the auto-encoder to integrate the information derived from texts. Let $x$ be the image, the CLIP textual encoder $c_\theta$ that maps the corresponding text $y$ into the latent space, the LDM loss is:

$$L_{LDM}(x,y) \coloneqq \mathbb{E}_{z \sim \mathcal{E}(x), \epsilon \sim \mathcal{N}(0,1), t} \left[ \| \epsilon - \epsilon_\theta \left( z_t, t, c_\theta(y) \right) \|_2^2 \right], \tag{1}$$

where $t$ denotes the time step, $z_t$ denotes the latent code noised at time step $t$, when $\epsilon, \epsilon_\theta$ represents the noised samples and the denoising U-Net [46], respectively. Through this noising-denoising procedure applied to the latent codes, LDM enables the underlying model to integrate information derived from texts into the visual results, while also allowing more flexibility to produce images.

**The OOD Concepts for LDMs.** Intuitively, out-of-distribution (OOD) concepts refer to the category of data whose distribution deviates significantly from what the model has learned. This degree of drifting can be quantified by an MMD score [27], which evaluates the discrepancy between ground-truth images and the generative results in the image latent space. Formally, for two probability distributions $P$ and $Q$, the MMD distance with respect to a positive definite kernel $k$ is:

$$\text{dist}_{MMD}^2(P,Q) \coloneqq \mathbb{E}_{x_p \sim P, x_p' \sim P} \left[ k(x,x') \right] + \mathbb{E}_{x_q \sim Q, x_q' \sim Q} \left[ k(x_q, x_q') \right] - 2\mathbb{E}_{x_p \sim P, x_q \sim Q} \left[ k(x_p, x_q) \right], \tag{2}$$

where $x_p, x_p'$ independently follow the distribution $P$ while $x_q, x_q'$ independently follow the distribution $Q$, with $k$ the Gaussian RBF kernel [17]. In our implementation, we sample two sets of vectors from the distribution $P$ and $Q$, then use CLIP embeddings [40] to calculate this score, which is also named as CMMD [27]. We set a concept with a high CMMD score (above 3.5) as an OOD concept.

**OOD Concepts are Hard to Adapt.** Recent text-to-image LDMs [45, 39] have achieved unprecedented performance on a wide range of concept tokens. However, we have found that there still exist many concepts that make LDMs fail after the adapter is fully trained. As we show in Figure 1, there are several discoveries: (i)-The concepts with higher CMMD scores are much more challenging for the underlying model to generate or adapt. The concepts with a CMMD score above 3.0 show explicit wrong visual details. For Axolotl and Frilled Lizards with CMMD above 4.0, the LDMs even generate the wrong species; (ii)-A higher CMMD score indicates a more severe loss of visual details when training adaptors. The concepts with CMMD scores above 3.5 in Axolotls and Emperor Penguin Chicks, show explicit distorted visual details, like color, texture, and delicate details. For Axolotl, the generative results show a wrong number and wrong positions of amateurs. For Emperor Penguin Chicks, the generative results show the wrong fur color of their heads and wings. To summarize, the higher the CMMD score of a concept, the more difficult it is for LDMs to adapt.

**The High-Quality Matters.** We further observe that the generative results are quite sensitive to training data when training adaptors meet OOD concepts. As shown in Figure 2, when the training images contain disruptive elements, the visual features of these disruptive elements will be easily introduced into the generative results. For example, when an adult emperor penguin appears in training data, then the black fur on their back can easily appear when generating their chicks, despite that their chicks have white fluffs as shown in the left part of Figure 2. If the object of the desired

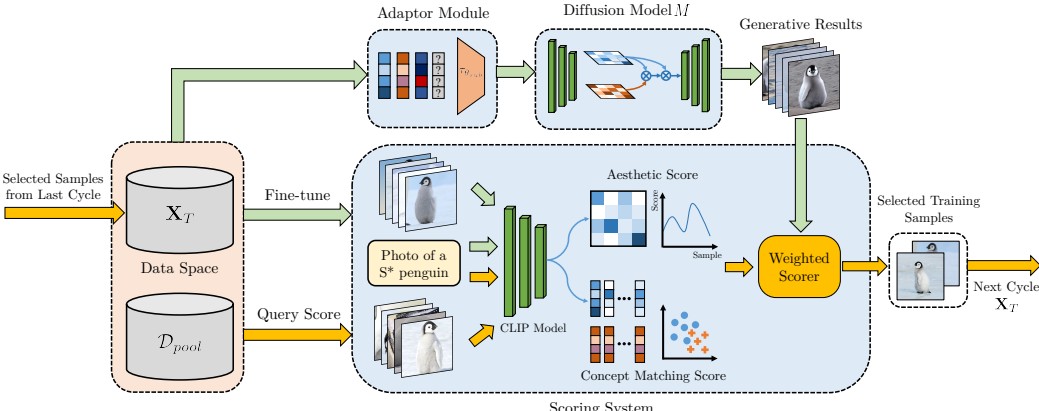

Figure 3: The overall pipeline of CATOD. In brief, CATOD alternatively performs data selection and scheduled OOD concept adaption. In each training cycle, we first generate OOD concepts according to the current adaptor and calculate the weights for the *aesthetic* score and *concept-matching* score. Then, we calculate the weighted score for each sample within the data pool $\mathcal{D}_{pool}$, select the top images accordingly, and add them to the training pool. At last, CATOD fine-tunes the scoring system and training adaptors according to the updated data pool, and proceed to the next cycle. The above three steps alternatively proceed until convergence.

concept appears to be too small within the image, then the generative results tend to be bad-looking, since the necessary visual details are not fully identified. In the middle part of Figure 2, we can see that distorted visual details like texture and shape in axolotl and emperor penguin chicks largely harm the aesthetics of the generated image. To generate images correctly and good-looking, we require enough amount of images with objects of high fidelity and do not contain disruptive elements, namely High-Quality samples as shown in the right part of Figure 2. Therefore, we aim to devise an effective data selection strategy for locating those high-quality images for these OOD concepts.

## 3 Method

### 3.1 Overall Architecture

We aim to adapt OOD concepts to LDMs correctly by an iterative data selection criterion that locates high-quality data and training adaptors accordingly as shown in Figure 3. Consequently, our method would alleviate issues introduced by disruptive elements, *e.g.*, irrelevant objects, and blurry images, while maintaining high fidelity of generative results. The core to CATOD is a scoring system, which consists of an aesthetic scorer and a concept-matching scorer, aiming to resolve the problems of incorrectly introduced visual details and distorted objects we have observed in Section 2. By properly trading off between these two scores, CATOD locates the most valuable samples for adapting underlying LDMs to OOD concepts and making the desired OOD concept depicted correctly.

### 3.2 The Scoring System

As described above, we need to estimate the potential impact of real-world samples over LDMs, according to which we select the most high-quality samples for training. In Section 2, we have observed two major factors that significantly impact generated image quality: object clarity and disruptive elements. Diving into the loss term $L_{LDM}(x, y)$ in Eq. (1), we may also observe that the training set $\mathbf{X}_T$ should be optimized towards both the underlying model (including the embedded adaptor) and the conditional text $y$, which indicates two important factors: object clarity and concept-matching achieved by preserving aesthetic information originally learned by LDMs and accurate image-text matching, respectively. Therefore, we attribute the quality of images to *aesthetic* and *concept-matching* and design two decoupled scorers accordingly.

**The Aesthetic Scorer.** Aesthetic evaluation is a long-standing field, which comprehensively considers whether the lighting, contrast, texture, and other photographic factors of an image are consistent with

human aesthetics. The general aesthetic scorer can be simply described as $p = S_{aes}(x)$, where $S_{aes}$ indicates the aesthetic scoring model, $p$ denotes the predicted score, and $x$ represents the input image. Following the work of PA-IAA [33], we fine-tune the generic aesthetic model and make it adapt to OOD concepts with personalized preference scores. In general, we assign a high score to the samples within the training set $\mathbf{X}_T$, assign a low score to samples from irrelevant categories and samples generated by underlying LDMs without an adaptor, and use them to fine-tune the aesthetic scorer. More details of this personalization are given in the Appendix B.3.

**The Concept-Matching Scorer.** Intuitively, concept-matching describes whether the OOD concepts get perfectly reflected in the generated results. A similar task is image retrieval, which is designed to retrieve images containing objects describing the desired concepts. Since image retrieval also relies on feature maps and some sort of matching score to retrieve images, we adopt matching score from VLAD-related works [28, 64, 55] as follows:

$$S_{con}(x) = \frac{1}{|\mathbf{X}_T|} \sum_{k=1}^{|\mathbf{X}_T|} \phi(r_x, r_k) \exp(\|r_x - r_k\|), \tag{3}$$

where $\phi(r_x, r_k) = 1$ if $r_k$ is the closet representation for $r_x$ and is set to 0 otherwise. Simply put, VLAD regards the extracted representations for $\mathbf{X}_T$ as a codebook and maps each image to its nearest code. Note that samples within $X_T$ mostly consist of clean and clear samples, this score is sufficient to quantify whether the object if exists in the given image distinguishes itself from other photographic components and matches the given concept.

### 3.3 Active Data Acquisition

**Optimization with Active Learning (AL).** Aiming to mitigate the problems caused by low-quality training samples, we proactively integrate the training data $\mathbf{X}_T$ into our objective as follows:

$$A^*, \mathbf{X}_T^* = \arg\min_{A, \mathbf{X}_T} \mathbb{E}_{x \sim \mathbf{X}_T} L_{LDM}(x, A, y). \tag{4}$$

Since the optimal set is initially unknown, a one-step optimization can easily lead to convergence to local optima. Therefore, we use an iterative paradigm to optimize adaptor $A$ and training data $\mathbf{X}_T$, respectively. In our implementation, we adopt the paradigm of AL [44, 54] to perform the optimization of $\mathbf{X}_T^{(t)}$ by data accumulation before training adaptors, with $t$ denotes the time step:

$$\mathcal{B}^{(t)} = \arg\min_{\mathcal{B} \subset \mathcal{D}_{pool} - \mathbf{X}_T^{(t-1)}, |\mathcal{B}| = b} \mathbb{E}_{x \sim \mathbf{X}_T^{(t-1)} \cup \mathcal{B}} L_{LDM}(x, A, y), \tag{5}$$

where $b$ is the number of samples added to the training pool at each cycle. The main reason for using AL is its preferred sample efficiency, with better controllable data bias management [15, 50]. Instead of repeatedly selecting data from the whole real-world data pool, AL provides a more efficient procedure to optimize training data by using data selection to accumulate high-quality training data. To this stage, the learning procedure of CATOD is relatively clear: the sample pool is progressively accumulated -by Eq. (5) -and the optimization of the adaptor $A$ is straightforward (Fig. 3).

Remember that AL involves iteratively updating the adaptor and the training data, it is important to design a training schedule for adaptors and determine how to acquire high quality based on the two scorers mentioned above. The primary objective of this design is to achieve a dynamic trade-off between the two scores. The specific details of these designs are described below.

**The Active Schedule for Training Adaptors.** The training schedule has been found crucial for successful adaption [47, 74, 65]. Since our training data continuously expands as the learning cycle of AL proceeds, the training schedule will be even more important. To arrange this schedule, we first calculate the aesthetics score $\gamma_{aes}(A) = \frac{1}{|g_A(\mathbf{X}_T)|} \sum_{x_g \in g_A(\mathbf{X}_T)} S_{aes}(x_g)$ and concept-matching score $\gamma_{con}(A) = \frac{1}{|g_A(\mathbf{X}_T)|} \sum_{x_g \in g_A(\mathbf{X}_T)} S_{con}(x_g)$ of the adaptor $A$ based on its generative results $g_A(\mathbf{X}_T)$, in which both the aesthetic score $\gamma_{aes}(A)$ and $\gamma_{con}(A)$ range from 0 to 10. Then, we use a trigonometric indicator that comprehensively measures its performance:

$$\gamma(A) = 10 \sin\left(\frac{\pi}{20}\gamma_{aes}(A)\right) \sin\left(\frac{\pi}{20}\gamma_{con}(A)\right). \tag{6}$$

Notably, $\gamma(A)$ peaks when the two scores get close or around the common value of 5.0 for most samples. Meanwhile, it bottoms when the scoring exhibits a stronger signal of being biased (to either

side). The properties emphasize that adapters balancing the two factors are of better quality. In training CATOD, we use $\gamma(A)$ as a signal to reduce the learning rate and stop training in time.

**Trading-off the Two Scores in Data Acquisition.** After getting the adaptor $A$, we continue to select the most suitable samples for the next-cycle training. Notice that whether newly selected images enhance the adaptor depends on both aesthetics and concept-matching, we ought to adjust our preferences according to the adaptor. In more detail, when the adaptor has high aesthetics but does not accurately depict the OOD concept, samples with high concept-matching scores better enhance the adaptor; when the adaptor fails to exhibit photographic attributes consistent with humans, samples with high aesthetics are preferred. To implement this preference, our acquisition score is:

$$S(x) = \left(1 - \sin\left(\frac{\pi}{20}\gamma_{aes}(A)\right)\right) S_{aes}(x) + \left(1 - \sin\left(\frac{\pi}{20}\gamma_{con}(A)\right)\right) S_{con}(x). \tag{7}$$

This formulation offers some meritable advantages. On one hand, the balancing terms are not pre-fixed or manually tuned, but dynamically dependent on the scores marginalized over the current generations. Further, when the score of either side gets larger, the corresponding balancing coefficient, in turn, decreases, thus attaining a proper trade-off. As a result, this mechanism encourages an alternation of sample selection towards both scoring metrics by selecting Top-$K$ samples to add to the training data $\mathbf{X}_T$ accordingly, which effectively guarantees the sample set diversity.

## 4 Theoretical Insights

This section presents our theoretical analyses of why aesthetic/concept-matching scores work in OOD Adaption. Specifically, we derive the distance between real-world data distributions and synthetic data distributions and then induce the important factors that affect this distance. We first introduce the *minimum mean square error* (MMSE) [22, 11, 6] to measure the discrepancy between distributions:

**Definition 4.1.** The *minimum mean square error (MMSE)* of estimating an input random vector $\widehat{\mathbf{X}} \in \mathbb{R}^n$ from an observation/output $\mathbf{X} \in \mathbb{R}^k$ is defined as

$$\mathrm{MMSE}(\widehat{\mathbf{X}}|\mathbf{X}) = \inf_{f \in \mathcal{M}(\mathbb{R}^n)} \mathbb{E}\left[\left\|\widehat{\mathbf{X}} - f(\mathbf{X})\right\|^2\right], \tag{8}$$

in which $\mathcal{M}(\mathbb{R}^n)$ denotes the space consisting of all measurable functions on $\mathbb{R}^n$.

Notice that we are trying to produce the best generation results which are initially unknown, our adaption task can be also regarded as estimating the optimal distribution with carefully selected data. By denoting the ideal generative results and the ideal adaptor with random variables $\widehat{\mathbf{X}}_G$ and $A^*$, respectively, we conclude that the LDM loss $L_{LDM}$ is consistent with this MMSE term:

**Theorem 4.2.** *Let $L_{LDM}$ be the LDM loss following Eq. 1, and the image space lies within $\mathbb{R}^n$. Then there always exists an ideal random vector $\widehat{\mathbf{X}}_G \in \mathbb{R}^n$ and an adaptor $A^*$ for LDM, such that*

$$\underset{\mathbf{X}_T \in \mathbb{R}^n}{\arg\min} \mathrm{MMSE}(\widehat{\mathbf{X}}_G|\mathbf{X}_T) = \underset{\mathbf{X}_T \in \mathbb{R}^n}{\arg\min} \mathbb{E}_{x \sim \mathbf{X}_T}\left[L_{LDM}(x, A^*)\right]. \tag{9}$$

Based on the preceding deduction, we have confirmed that the change in MMSE can also indicate the change in LDMs. To dive deep into the MMSE term, we further decompose it as follows:

**Theorem 4.3.** *(Pythagorean Theorem for MMSE [13].) Following Theorem 4.2, by setting $f$ to the generative model $g_A$, the MMSE term in Eq. (8) can be decomposed into two terms as follows:*

$$\mathbb{E}\left[\left\|\widehat{\mathbf{X}}_G - g_A(\mathbf{X}_T)\right\|\right] = \mathbb{E}\left[\left\|\widehat{\mathbf{X}}_G - g_{A^*}(\mathbf{X}_T)\right\|\right] + \mathbb{E}\left[\left\|g_{A^*}(\mathbf{X}_T) - g_A(\mathbf{X}_T)\right\|\right], \tag{10}$$

in which $A^*$ denotes a potential ideal adaptor containing information for text $y$. Notice that the generative results $\mathbf{X}_G$ relies on the training set $\mathbf{X}_T$, $g_A(\mathbf{X}_T)$ can also be written as $\mathbf{X}_G|\mathbf{X}_T$.

Upon revisiting the two terms, we can gain a deeper understanding of the relationship between MMSE and aesthetic/concept-matching score: (i)-The first term is focused on estimating the difference between the generative distribution $\mathbf{X}_G|\mathbf{X}_T$ and the ideal distribution of $\widehat{\mathbf{X}}_G$. This assessment helps

Table 1: A Comparison over the performance of CATOD, in terms of the CLIP score and CMMD score with 100 images sampled at last. This table shows the average result of 5 sub-classes within each category. The overall improvement of our proposed CATOD is provided by "Imp.". Methods with the best performance are bold-folded.

| Comparison Methods | CLIP Score (↑) | | | | | | | CMMD [27] (↓) | | | | | | |
|---|---|---|---|---|---|---|---|---|---|---|---|---|---|---|
| | insect | lizard | penguin | seafish | snake | Avg. | Imp. | insect | lizard | penguin | seafish | snake | Avg. | Imp. |
| DreamBooth[47] + RAND | 65.35 | 67.89 | 68.07 | 65.59 | 72.07 | 67.79 | ⇑7.58 | 1.35 | 1.39 | 1.67 | 1.59 | 1.25 | 1.45 | ⇓0.50 |
| DreamBooth[47] + CLIP | 70.56 | 72.19 | 72.09 | 71.17 | 74.59 | 72.12 | ⇑3.25 | 1.04 | 1.24 | 1.05 | 1.38 | 1.16 | 1.17 | ⇓0.22 |
| DreamBooth[47] + CATOD | **72.18** | **75.30** | **75.16** | **74.60** | **79.61** | **75.37** | - | **0.92** | **1.08** | **0.80** | **1.21** | **0.74** | **0.95** | - |
| TI[18] + RAND | 58.24 | 59.34 | 63.45 | 61.35 | 62.34 | 60.94 | ⇑7.04 | 2.45 | 2.15 | 2.09 | 1.95 | 1.65 | 2.06 | ⇓0.63 |
| TI[18] + CLIP | 61.65 | 67.24 | 66.95 | 62.27 | 64.23 | 64.47 | ⇑3.51 | 2.11 | 1.86 | 1.44 | 1.87 | 1.37 | 1.73 | ⇓0.30 |
| TI[18] + CATOD | **69.21** | **70.14** | **68.23** | **64.89** | **67.41** | **67.98** | - | **1.57** | **1.73** | **1.15** | **1.43** | **1.25** | **1.43** | - |
| LoRA[25] + RAND | 64.39 | 65.02 | 67.49 | 68.87 | 71.09 | 67.37 | ⇑10.60 | 1.52 | 1.47 | 1.56 | 1.61 | 1.18 | 1.47 | ⇓0.63 |
| LoRA[25] + CLIP | 70.27 | 74.13 | 72.08 | 73.19 | 75.64 | 73.06 | ⇑4.91 | 1.29 | 1.33 | 1.04 | 1.35 | 0.89 | 1.18 | ⇓0.34 |
| LoRA[25] + CATOD | **72.60** | **77.00** | **74.11** | **84.29** | **81.86** | **77.97** | - | **0.94** | **0.89** | **0.71** | **0.88** | **0.77** | **0.84** | - |

us understand how the introduced samples lose visual information within the underlying LDM, thus we can connect this term to aesthetic preservation, i.e. aesthetic score; (ii)-The second term illustrates the degree to which the training set $\mathbf{X}_T$ distorts the OOD concept information within the ideal adaptor. Hence, a concept-matching score accurately portrays how newly given samples impact this term. At this stage, we have completed the theoretical support showing that both aesthetics and concept-matching are major factors that influence the performance of adaptors.

## 5 Experiments

In this section, we present the main experimental results both qualitatively and quantitatively. To evaluate our proposed CATOD, we combine it with several works for adaption, i.e. DreamBooth [47], Textual Inversion [18], and LoRA [25]. More experimental results can be found in the Appendix. *The source code is attached in the Supplementary.*

**Datasets.** We test our method on datasets with 25 OOD concepts that can hardly be generated through prompt engineering on the text-to-image model. This dataset consists of 5 categories: insect, lizard, penguin, seafish, and snake, and each category contains data from 5 OOD concepts. Each concept has 1,000 examples in total with 100 samples left out for validation. The dataset is collected from publicly available datasets including ImageNet, iNaturalist 2018 [67], IP102 [71].

**Implementation Details.** We conduct the active generation experiments on our proposed CATOD and three representative adaptors, i.e. DreamBooth [47], Textual Inversion [18] (termed as TI in the paper), and LoRA [25]. Since there are currently no available studies that focus on locating "high-quality" samples for training, we apply random sampling (RAND), and CLIP-score-based sampling (CLIP) for each baseline in our active learning setting. Each experiment starts with 20 randomly sampled instances, and we conducted 5 cycles of data accumulation in which we selected 20 "good" samples to add to the training pool. We train 20 epochs for all combinations of adaption techniques and sampling strategies in each active learning cycle, with a batch size of 1. Furthermore, we generate 100 images for each concept for evaluation. We use the commonly adopted Stable Diffusion 2.0 pre-trained on LAION-5B [51] following Rombach's work [45].

**Evaluation Metrics.** We evaluate the quality of our generated images with the widely used CLIP score [20] and the recently proposed CMMD score [27], which quantify the model performance in two aspects. Specifically, the CLIP score measures how generated images match the given text, which is expected to be as high as possible. Meanwhile, the CMMD score evaluates the discrepancy between generated images with the real ones, indicating better generative results with lower values.

### 5.1 Single-concept Generation Results

To evaluate the performance of CATOD on OOD concepts, we test it on all 25 target concepts one by one separately and report the average performance of 5 concepts within each category in Table 1. We show the superior results of our CATOD with respect to both qualitative and quantitative comparisons.

**Qualitative Comparisons.** We qualitatively compare CATOD with other sampling strategies according to their generated images based on LoRA [25]. With random sampling (RAND), we observe that the generative results only partially learned some photographic attributes like color and texture,

Table 2: A Comparison over the performance of CATOD when training with images from multiple concepts, in terms of the CLIP score and CMMD score with 100 images sampled at last. In each experiment, we sample images from all the sub-classes within each category and check whether the fine-tuned model can generate all 5 concepts. The overall improvement of our proposed CATOD is provided by "Imp.". Methods with the best performance are bold-folded.

| Comparison Methods | CLIP Score (↑) | | | | | | | CMMD [27] (↓) | | | | | | |
| --- | --- | --- | --- | --- | --- | --- | --- | --- | --- | --- | --- | --- | --- | --- |
| | insect | lizard | penguin | seafish | snake | Avg. | Imp. | insect | lizard | penguin | seafish | snake | Avg. | Imp. |
| DreamBooth[47] + RAND | 63.29 | 63.79 | 65.72 | 65.59 | 64.36 | 64.55 | ⇑8.20 | 1.74 | 2.14 | 2.16 | 2.02 | 1.85 | 1.98 | ⇓0.75 |
| DreamBooth[47] + CLIP | 67.57 | 70.35 | 71.10 | 69.34 | 70.38 | 69.75 | ⇑3.00 | 1.53 | 1.76 | 1.80 | 1.79 | 1.59 | 1.69 | ⇓0.46 |
| DreamBooth[47] + CATOD | **70.83** | **72.28** | **74.31** | **70.90** | **75.45** | **72.75** | - | **1.39** | **1.25** | **1.34** | **1.45** | **0.73** | **1.23** | - |
| TI[18] + RAND | 59.23 | 56.97 | 57.90 | 60.83 | 62.65 | 59.52 | ⇑5.88 | 2.84 | 2.56 | 2.27 | 2.39 | 2.41 | 2.49 | ⇓0.83 |
| TI[18] + CLIP | 61.74 | 60.72 | 63.79 | 62.71 | 65.71 | 62.93 | ⇑2.47 | 2.26 | 2.08 | 1.94 | 1.97 | 2.23 | 2.10 | ⇓0.44 |
| TI[18] + CATOD | **64.48** | **63.93** | **65.93** | **65.44** | **67.24** | **65.40** | - | **1.76** | **1.53** | **1.65** | **1.64** | **1.73** | **1.66** | - |
| LoRA[25] + RAND | 63.85 | 65.27 | 66.46 | 68.25 | 69.43 | 66.65 | ⇑7.64 | 1.79 | 2.05 | 1.93 | 1.85 | 1.55 | 1.83 | ⇓0.59 |
| LoRA[25] + CLIP | 69.25 | 70.84 | 71.39 | 71.91 | 72.78 | 71.23 | ⇑3.06 | 1.40 | 1.69 | 1.74 | 1.59 | 1.28 | 1.54 | ⇓0.30 |
| LoRA[25] + CATOD | **71.19** | **74.09** | **73.68** | **75.60** | **76.90** | **74.29** | - | **1.13** | **1.37** | **1.49** | **1.26** | **0.95** | **1.24** | - |

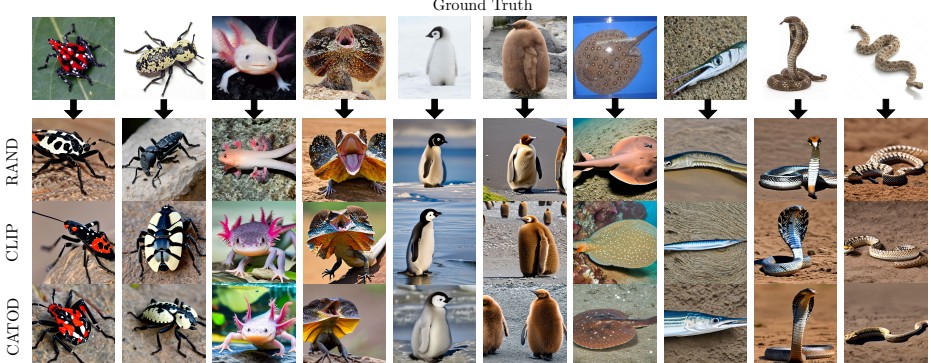

Figure 4: **A comparison of different sampling strategies with LoRA.** Specifically, we compare three lines of works: (1) RAND, in which the model is trained with 100 randomly selected samples; (2) with samples of the highest CLIP scores (100 samples); (3) 100 samples with CATOD. The model trained with randomly sampled data fails to capture the features of out-of-distribution (OOD) concepts, while the ones trained with top CLIP scores contain necessary details but also include disruptive elements.

but failed to make the objects have the correct appearance and shape. CLIP-based sampling (CLIP) somehow produces the corrected shape of the concept but still fails to capture the necessary details for describing the object. In comparison, our proposed CATOD successfully matches all the photographic attributes, while also guaranteeing the image aesthetics.

To further look at how CATOD learns the photographic attributes that precisely match the concept, we show samples generated from different cycles in Fig. 7. We can see that some attributes like color, and texture are already learned in cycle 1, but the shape of the object does not match the ground-truth ones. From cycle 1 to 3, CATOD shows a clear shape modification, making the objects more like the real ones. From cycles 3 to 5, an iterative refinement on more photographic details like light, contrast, and other minor modifications (like the beak for penguins and antenna for axolotls) is shown in generative results, making them hard to distinguish from the ground-truth ones even with a careful look. At cycle 5 and later cycles, the image quality stabilizes and we can hardly see enhancement apart from image diversity. To conclude, we can see an explicit attribute matching process from easy ones to the finer ones within CATOD, showing the importance and effectiveness of iterative training.

**Quantitative Comparisons.** Table 1 reports CLIP scores [20] and CMMD scores [27] from each strategy with models trained on 25 different OOD concepts and evaluated through the generative results. Specifically, we evaluate the performance of CATOD on each concept and average the results within each category in Table 1. We have the following conclusions: (1) the average performance of our strategies outperforms all compared methods by a 0.56∼11.10 CLIP score, justifying the effectiveness of locating aesthetic and clean samples over text-image matching. (2) CATOD also brings a 0.12∼0.54 CMMD decrease on various frameworks and concepts, indicating better image alignment with proper sampling guided by CATOD. To summarize, both image-matching and text-matching scores exhibit better results compared to the baselines, suggesting that our proposed CATOD significantly outperforms other strategies, which is consistent with the qualitative results in Fig. 4.

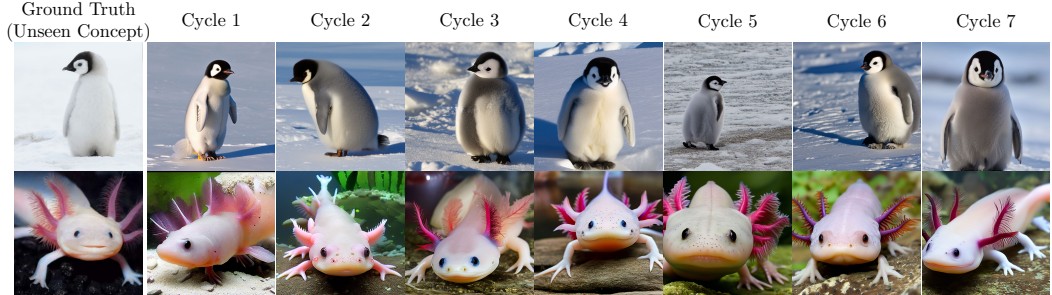

| Ground Truth (Unseen Concept) | Cycle 1 | Cycle 2 | Cycle 3 | Cycle 4 | Cycle 5 | Cycle 6 | Cycle 7 |

Figure 5: **Generative results as cycle proceeds**. Samples are generated with CATOD on cycles from 1 to 7. To better observe how generated images change as the cycle proceeds, we conduct another 2 cycles here. In each cycle, we select and add 20 high-quality samples. Generative samples start to converge and contain the right details within the original concept after cycle 4 or 5. We can also see that those generative results contain diverse contents within the background based on the few images given.

Table 3: **Results of Ablating Aesthetic, Concept-Matching Scorer and Weighted Scoring on CATOD.** We show the average results conducted on the categories "penguin" and "lizard" with LoRA.

| Modules | | | CLIP(↑) | | CMMD(↓) | |
|---|---|---|---|---|---|---|
| Aesthetic | Concept Matching | Weighted Scoring | lizard | penguin | lizard | penguin |
| ✓ | | | 73.19 | 72.85 | 1.14 | 1.28 |
| | ✓ | | 68.45 | 70.16 | 1.10 | 1.32 |
| ✓ | ✓ | | 75.35 | 73.24 | 0.94 | 0.93 |
| ✓ | ✓ | ✓ | **77.00** | **74.11** | **0.89** | **0.71** |

Table 4: **Results of CATOD with different types of aesthetic scorers.** We show the average results conducted on the categories "penguin" and "lizard" with LoRA.

| Type of Aesthetic Score | CLIP(↑) | | CMMD(↓) | |
|---|---|---|---|---|
| | lizard | penguin | lizard | penguin |
| ReLIC [81] | 71.35 | 72.39 | 1.29 | 1.59 |
| TANet [23] | 72.05 | 72.67 | 1.32 | 1.35 |
| BAID [77] | 73.08 | 72.79 | 1.18 | 1.37 |
| Ours | **73.19** | **72.85** | **1.14** | **1.28** |

## 5.2 Multi-concept adaption Results.

To further investigate whether CATOD could adapt multiple concepts simultaneously, we group the 25 concepts by category and train the adaptation model on each category. To be specific, we compare our baselines using a generated set consisting of 500 images (100 images per concept) and exhibit our results in Table 2. Following the setting of single-concept adaption, we conduct 10 cycles of data accumulation and get 200 samples at last, since multi-concept adaption requires more data. We still observe a notable performance gain with a 1.56~5.07 CLIP score increase and a 0.12~0.86 CMMD score decrease compared to baselines. These results verify that our CATOD still achieves better performance on multi-concept adaption.

## 5.3 Ablation Studies

We verify the efficacy of all components in our proposed CATOD in Table 3 and 4, including the aesthetical scoring module and the concept-matching mechanism within the weighted scoring system.

**W/O Aesthetic Score.** First, we validate the effectiveness of CATOD by removing the aesthetic scores. A significant decrease in performance (up to 8.55) on the CLIP score can be observed in Table 3. We attribute this decline to the fact that matching-based metrics prioritize image representations over the given concept, leading to a deterioration in image-text matching.

**The Type of Aesthetic Scores.** To further investigate the impact of aesthetic scores, we have also applied different types of aesthetics with CATOD in Table 4. We observed that recent state-of-the-art aesthetic evaluations did not improve OOD adaption and even led to minor performance loss. This could be attributed to the fact that these models were originally designed for general aesthetic assessments, whereas our aesthetic scorer is highly personalized towards specific OOD concepts.

**W/O Concept-Matching Score.** After removing the concept-matching scorer, we observed that the adaptor tends to perform better compared to just removing the aesthetic scorers. This might be due to the aesthetic scorer being designed based on CLIP backbones, showing some consistency with the CLIP score. However, it still demonstrates limitations based on the image-matching score CMMD. While these aesthetic qualities partly describe the clarity and accuracy of the object, they do not adequately focus on the image representation space.

**The Weighted Score.** We have confirmed the effectiveness of balancing two scores by simply adding them together to guide CATOD (Table 3, Line 5). We observed that this resulted in consistent performance loss for both the CLIP score (up to 1.65) and the CMMD score (up to 0.22). This suggests that both text-image matching and image-to-image matching are affected by the trade-off between aesthetics and concept matching, further emphasizing the importance of these two scores.

# 6  Related work

**Personalized Text-to-image Synthesis with Adaptors.** The task of text-to-image generation involves creating specific images based on text descriptions [3, 73, 79, 4], and has achieved impressive performance with state-of-the-art diffusion models [41, 45, 39]. Therefore, *adapting* large-scale text-to-image models to a specific concept while also preserving this amazing performance, i.e. *personalization* [7], has become another recent research interest. But this is often difficult since re-training a model with an expanded dataset for each new concept is prohibitively expensive while fine-tuning the whole model [12, 32] or transformation modules [83, 20, 60] on few examples typically leads to some rate of forgetting [30]. Therefore, *adaptors* such as Textual Inversion [18], DreamBooth [47, 48, 5, 58], LoRA [25, 66, 86, 76], along with some other works [70, 19] derived from them, have become more commonly adopted. Typically, they focus on a small but crucial part of the model or extra networks inserted into underlying models, thus more computationally-efficient, while also preserving the efficacy of the underlying models with lower computational costs. For example, *textual inversion* (TI) [18] represents the newly-given concept with pseudo word [42] and remapping it to another carefully trained embedding in the text-encoding space, guided by few images. Despite their computational efficiency, these approaches are still facing difficulties dealing with out-of-distribution concepts as we observe in Section 2.

**Active Learning and Selection.** Active Learning is a machine learning paradigm that involves actively selecting the most suitable data for training models from external data sources [44, 63]. The most crucial part of active learning is the strategy to locate the optimal data batch. Current studies can be roughly categorized as follows: (a) Score-based methods that prefer the samples with the highest information scores [36, 69, 10]; (b) Representation-based methods searching for the samples that are the most representative of the underlying data distribution [53, 1, 62]. The Active Selection paradigm within Active Learning serves as an efficient and powerful dataset curation tool (i.e. *adaption*), leading to numerous studies adopting this paradigm from a wide range of subjects [68, 24, 8]. Meanwhile, due to the effectiveness of Active Learning in selecting the most suitable training samples, recent studies have utilized similar sampling strategies to address challenges associated with long-tailed distributions [56, 57] and noisy data [72]. Given that Out-Of-Distribution (OOD) concepts in Latent Diffusion Models (LDMs) often involve unseen or long-tailed concept tokens [79], this motivates us to leverage Active Learning for selecting samples that are well-suited for training adaptors.

In this work, we focus on scored-based strategies in two-fold: *aesthetics* and *concept-matching*. Image Aesthetics Assessment (IAA) aims at evaluating image aesthetics computationally and automatically [9], while automatically assessing image aesthetics is useful for many applications [35, 29, 14]. To take a step further, personalized image aesthetics assessment (PIAA) [43, 85, 75] was proposed to capture unique aesthetic preferences, consistent with our goal to adjust our paradigms accordingly different concepts. At the same time, "Concept-matching" is adopted from the field of image retrieval, in which we search for relevant images in an image gallery by analyzing the visual content (e.g., objects, colors, textures, shapes *etc*.), given a query image [61, 31]. To design a paradigm that automatically meets the harsh requirements for adaptor training, we consider both two factors.

# 7  Conclusion

We propose CATOD, an enhanced, data-efficient, and practically useful version of the OOD concept adaptation for AIGC. This method is encapsulated in an active-learned paradigm with carefully designed acquisitional scoring mechanisms. CATOD significantly outperforms the prior approaches in many (if not most) aspects including generation quality, concept matches, technological robustness, data efficiency, etc. In the future, we hope to ship CATOD to the open-source community so as to absorb more OOD concepts that were originally uncovered.

## Acknowledgements

This work is supported by the Pioneer R&D Program of Zhejiang (No. 2024C01035). This work is also partially supported by the Fundamental Research Funds for the Central Universities (226-2024-00049) and (226-2024-00145).

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

# A Theory

This section provides a complete derivation for the analysis given in Section 4. In brief, we first link $L_{LDM}$ to a *minimum mean square error* (MMSE) term in Theorem A.2 (i.e. Theorem 3.2 within the main context), then decompose the MMSE term to see how we minimize MMSE in different aspects in Theorem A.3 (i.e. Theorem 3.3 within the main context) in the main context. For convenience, we reclaim some of the formulations at the beginning of this section:

**Definition A.1.** The *minimum mean square error (MMSE)* of estimating an input random vector $\widehat{\mathbf{X}} \in \mathbb{R}^n$ from an observation/output $\mathbf{X} \in \mathbb{R}^k$ is defined as

$$\text{MMSE}(\widehat{\mathbf{X}}|\mathbf{X}) = \inf_{f \in \mathcal{M}(\mathbb{R}^n)} \mathbb{E}\left[\left\|\widehat{\mathbf{X}} - f(\mathbf{X})\right\|^2\right], \tag{11}$$

in which $\mathcal{M}(\mathbb{R}^n)$ denotes the space consisting of all measurable functions on $\mathbb{R}^n$.

Based on Eq. (1), we can adapt the LDM to an arbitrary concept $y$ with a corresponding image set $X$ that describes this concept. Typical adaptations are mostly based on fine-tuning all the parameters $\theta$, which is quite costly and requires a lot of data. However, recent studies have found that training only a small part of the model [18, 47] or inserting extra networks [25, 26] could attain the same performance, which largely alleviates the computational burden with far fewer samples need for training. We call this part of parameters or networks as "*adaptors*" and denote them by $A$. Then, the fine-tuning process of the *adaptors*, also named *adaption*, can be formulated as follows:

$$A^* = \arg\min_A \mathbb{E}_{x \sim X, z \sim \mathcal{E}(x), \epsilon \sim \mathcal{N}(0,1), t}\left[\|\epsilon - \epsilon_{\theta, A}(z_t, t, c_{\theta, A}(y))\|_2^2\right] \tag{12}$$

Therefore, $v$ be a conditioning vector corresponding to some given text $y$, the LDM loss is:

$$L_{LDM}(x, A, y) \coloneqq \mathbb{E}_{x \sim X, z \sim \mathcal{E}(x), \epsilon \sim \mathcal{N}(0,1), t}\left[\|\epsilon - \epsilon_{\theta, A}(z_t, t, c_{\theta, A}(y))\|_2^2\right] \tag{13}$$

Following the definition of LDM loss, we continue to link LDM loss to the MMSE term in the theorem as follows (in which we omit the parameter $y$ since it is independent of other parameters):

**Theorem A.2.** *Let $L_{LDM}$ be the LDM loss following Eq.* (13)*, and the image space lies within $\mathbb{R}^n$. Then there always exists an ideal random vector $\widehat{\mathbf{X}}_G \in \mathbb{R}^n$ and a condition vector $v^*$ within the text embedding space for LDM, such that*

$$\arg\min_{\mathbf{X}_T \in \mathbb{R}^n} \text{MMSE}(\widehat{\mathbf{X}}_G|\mathbf{X}_T) = \arg\min_{\mathbf{X}_T \in \mathbb{R}^n} \mathbb{E}_{x \sim \mathbf{X}_T}[L_{LDM}(x, A^*)].$$

*Proof.* By denoting an ideal generative distribution with $\widehat{\mathbf{X}}_G$, which is produced by the generative model with a minimized LDM loss, we continue our proof. Set $f_A(\mathbf{X}_T) = \mathbb{E}_{x \sim \mathbf{X}_T}[x, A]$, we can see that $f_A$ is also a functional controlled by the adaptor $A$. Following the definition of MMSE and a fixed $\mathbf{X}_T$, we set $A^*$ as:

$$A^* = \arg\min_v \mathbb{E}_{x \sim \mathbf{X}_T}[L_{LDM}(x, A)]. \tag{14}$$

With this $A^*$, and the Universal Approximation Theorem [38] for all deep learning models, $f_{A^*}$ becomes the functional needed to quantify the MMSE term following its definition. Therefore, with proper $\mathbf{X}_G, A^*$, we have

$$\text{MMSE}(\widehat{\mathbf{X}}_G|\mathbf{X}_T) = \mathbb{E}_{x \sim \mathbf{X}_T}[L_{LDM}(x, A^*)],$$

thus completing the proof of this theorem. $\square$

**Theorem A.3.** *(Pythagorean Theorem for MMSE [13].) Following Theorem A.2, by setting $f$ to the generative model $g_{A^*}$ with an ideal adaptor $A^*$ containing sufficient OOD concept information, rewrite $g_A(\mathbf{X}_T)$ to $\mathbf{X}_G|\mathbf{X}_T$, and the minimum mean square error (MMSE) in Eq.* (11) *can be decomposed into two terms:*

$$\mathbb{E}\left[\left\|\widehat{\mathbf{X}}_G - \mathbb{E}[\mathbf{X}_G|\mathbf{X}_T]\right\|\right] = \mathbb{E}\left[\left\|\widehat{\mathbf{X}}_G - g_{A^*}(\mathbf{X}_T)\right\|\right] + \mathbb{E}[\|g_{A^*}(\mathbf{X}_T) - \mathbb{E}[\mathbf{X}_G|\mathbf{X}_T]\|], \tag{15}$$

*Proof.* This equation can be further decomposed into two terms:

$$\mathbb{E}\left[\left(\widehat{\mathbf{X}}_G - \mathbb{E}\left[\mathbf{X}_G | \mathbf{X}_T\right]\right)\phi(X_T)\right]$$
$$= \mathbb{E}\left[(\widehat{\mathbf{X}}_G - \mathbf{X}_G)\phi(\mathbf{X}_T)\right] + \mathbb{E}\left[\mathbf{X}_G - \mathbb{E}\left[\mathbf{X}_G | \mathbf{X}_T\right]\phi(\mathbf{X}_T)\right].$$

Since $\widehat{\mathbf{X}}_G - \mathbf{X}_G$ represents the variance for generative results and is orthogonal to $\mathbf{X}_T$, the first term is 0, making it sufficient to consider only the second term. To prove that the second term is also 0, we first prove the orthogonality property, i,e.

$$\mathbb{E}\left[\left(\widehat{\mathbf{X}}_G - \mathbb{E}\left[\mathbf{X}_G | \mathbf{X}_T\right]\right)\phi(X_T)\right] = 0 \tag{16}$$

for any function $\phi$.

$$\mathbb{E}\left[\mathbb{E}\left[\mathbf{X}_G | \mathbf{X}_T\right]\phi(\mathbf{X}_T)\right]$$
$$= \sum_{x_t} \mathbb{E}\left[\mathbf{X}_G | \mathbf{X}_T = x_t\right]\phi(x_t)P(\mathbf{X}_T = x_t)$$
$$= \sum_{x_g}\left[\sum_{x_t}\frac{P(\mathbf{X}_G = x_g, \mathbf{X}_T = x_t)}{P(\mathbf{X}_T = x_t)}\right]\phi(x_t)P(\mathbf{X}_T = x_t)$$
$$= \sum_{x_g}\sum_{x_t} x_g\phi(x_t)P(\mathbf{X}_G = x_g, \mathbf{X}_T = x_t)$$
$$= \mathbb{E}\left[\mathbf{X}_G\phi(\mathbf{X}_T)\right]$$

Therefore, we have the orthogonal property in Eq. (16).

Now we continue our proof for the main theorem. First, we can decompose Eq. (15) as follows:

$$\mathbb{E}\left[\left\|\widehat{\mathbf{X}}_G - \mathbb{E}\left[\mathbf{X}_G | \mathbf{X}_T\right]\right\|^2\right]$$
$$= \mathbb{E}\left[\|\widehat{\mathbf{X}}_G - g_{A^*}(\mathbf{X}_T) + g_{A^*}(\mathbf{X}_T) - \mathbb{E}\left[\mathbf{X}_G | \mathbf{X}_T\right]\|^2\right]$$
$$= \mathbb{E}\left[\|\widehat{\mathbf{X}}_G - g_{A^*}(\mathbf{X}_T)\|^2\right] + \mathbb{E}\left[g_{A^*}(\mathbf{X}_T) - \mathbb{E}\left[\mathbf{X}_G | \mathbf{X}_T\right]\|\right]$$
$$+ 2\mathbb{E}\left(\widehat{\mathbf{X}}_G - g_{A^*}(\mathbf{X}_T)\right)\left(g_{A^*}(\mathbf{X}_T) - \mathbb{E}\left[\mathbf{X}_G | \mathbf{X}_T\right]\right)$$

Since the gap between pure generative result guided by the training set $\mathbf{X}_T$ and an ideal text embedding is purely influenced by the training set $\mathbf{X}_T$, $g_{A^*}(\mathbf{X}_T) - \mathbb{E}\left[\mathbf{X}_G | \mathbf{X}_T\right]$ can also be regarded as a functional over $\mathbf{X}_T$, making the last term equal to 0 according to the orthogonal property in Eq. (16). At this step, we obtain the result in Eq. (15). □

## B  More Details About CATOD.

In this section, we provide more details about our proposed Controllable Adaptor Towards Out-of-Distribution Concepts (CATOD).

### B.1  An Overall Paradigm

Initially, we have a large data pool $D_{pool}$ related to the concept given with different quality and begin with an initial randomly-sampled training dataset $\mathbf{X}_T^{(0)}$. $\mathbf{X}_T$ might contain distorted images or even mismatch the given concept, which is common in publicly available datasets. Following the typical paradigm of Active Learning, we train all the models we use on the training set $\mathbf{X}_T$. Then, we select a batch of data $\mathcal{B}$ from the data pool $D_{pool}$ according to the acquisition functions induced from models. Finally, we move this selected batch of data $\mathcal{B}$ to $\mathbf{X}_T$, go back to the model training step to re-train or adapt the models, and repeat this cycle until $\mathbf{X}_T$ reaches its maximum capacity or the quality of models cannot be further enhanced by adding new data. An overall Algorithm for a cycle of CATOD is provided in Alg. 1.

---

**Algorithm 1** An active selection cycle for CATOD.

---

**Input**: Text-to-image model $g_A(\cdot)$ that can be integrated with adaptor $A$, real-world data pool $D_{pool}$, training data pool $\mathbf{X}_T$, budget $b$, pretrained aesthetic scorer $S_{aes}(\cdot)$, concept-matching scorer $S_{con}(\cdot)$, adaptor $A$, learning rate group $R$.
**Output**: Updated training pool $\mathbf{X}_T$, updated adaptor $A$.

1: Fine-tune the aesthetic scorer $S_{aes}(\cdot, \theta)$ on $\mathbf{X}_T$ following Sec.B.3;
2: Calculate the aesthetic score $\gamma_{aes}(A) = \frac{1}{|g_A(\mathbf{X}_T)|} \sum_{x_g \in g_A(\mathbf{X}_T)} S_{aes}(x_g)$ and concept-matching
    score $\gamma_{con}(A) = \frac{1}{|g_A(\mathbf{X}_T)|} \sum_{x_g \in g_A(\mathbf{X}_T)} S_{con}(x_g)$ for $A$;
3: **for** $x \in D_{pool}$ **do**
4:     Calculate the integrated score $S(x)$(Eq.(7));
5: **end for**
6: $\mathcal{B} \leftarrow \{$Top-b samples within $D_{pool}$ according to $S\}$;
7: $s \leftarrow \gamma(A)$ (Eq.(6));
8: $r \leftarrow$ the largest value within $R$;
9: **while** $r \neq \min(R)$ **do**
10:     $A' \leftarrow A - r\nabla_A \sum_{x \in \mathbf{X}_T \cup \mathcal{B}} L_{LDM}(x, A)$;
11:     Generate a small-scale $\mathbf{X}_G$ with $g(\cdot)$ conditioned on $A'$;
12:     Calculate score $\gamma(A)$ for $A$ via Eq.(6)) and $\mathbf{X}_G$;
13:     **if** $\gamma(A) > s$ **then**
14:         $A \leftarrow A'$;
15:         $s \leftarrow \gamma(A)$;
16:     **else**
17:         $r \leftarrow$ a smaller value than $r$ within $R$
18:     **end if**
19: **end while**
20: $\mathbf{X}_T \leftarrow \mathbf{X}_T \cup \mathcal{B}$

---

We use a weighted-scoring system as the acquisition module that evaluates the quality of images and adaptors in our proposed CATOD framework. In traditional active learning or data selection, the acquisition module is only conducted on samples. However, we observed that an increase in the number of samples does not necessarily contribute to a performance boost like in a traditional active learning setting (which we will verify in Appendix C). Therefore, we propose this weighted scoring system to help schedule adaptor training, which helps locate the best version among them and select samples according to their quality.

Our Controllable Adaptor Towards Out-of-Distribution Concepts framework can be divided into 4 steps: (1) Train the scoring system on the current training set $\mathbf{X}_T$. (2) Train the adaptor on the training set $\mathbf{X}_T$ and schedule the training accordingly to the scoring system; (3) Calculate the acquisition score accordingly to the scoring system and the best adaptor for samples in dataset $D_{pool}$; (4) Select top-K samples from $D_{pool}$ and move them to train set $\mathbf{X}_T$.

## B.2 An Implementation of MMD score.

To give out an quantifiable and unbiased estimation of Eq.(2), we sample two sets of vectors $X = \{x_1, x_2, \ldots, x_m\}$ from $P$ and $G = \{g_1, g_2, \ldots, g_n\}$ from $Q$, an estimation can be give by:

$$\text{dist}^2_{MMD}(X, G) := \frac{1}{m(m-1)} \sum_{i=1}^{m} \sum_{j \neq i}^{m} k(x_i, x_j) + \frac{1}{n(n-1)} \sum_{i=1}^{n} \sum_{j \neq i}^{n} k(g_i, g_j) - \frac{2}{mn} \sum_{i=1}^{m} \sum_{j=1}^{n} k(x_i, g_j). \tag{17}$$

## B.3 Aesthetic Scorer

Following the description of the aesthetic scorer in Section 3.2, we explain how to personalize it here. Since our primary goal is to make the model distinguish low-quality/high-quality examples for OOD concepts, we design an automatic procedure to assign aesthetic scores to our training set $\mathbf{X}_T$ at each cycle, based on which we attain an aesthetic training set $\mathbf{X}_T^{aes}$. In this work, the aesthetics training comprises three parts: (1) The original training set $\mathbf{X}_T$; (2) Generated set $D_{TG}$ without assistance

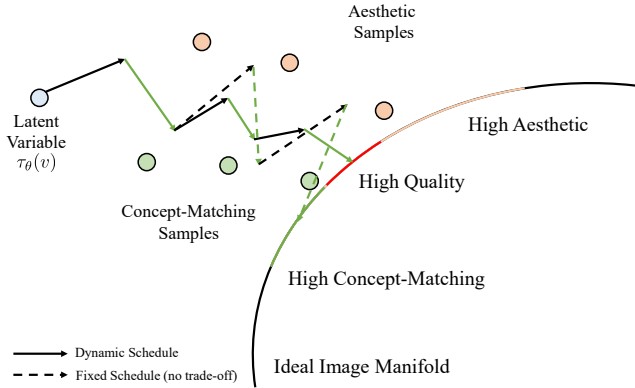

Figure 6: **An intuitive comparison for fixed/dynamic schedules.** The active learning paradigm can be viewed as guiding the iterative embedding updating through newly added samples. We can see that a fixed schedule makes the learned embedding heavily biased, which in turn leads to performance fluctuation.

from any adaptors; (3) Randomly sampled set $D_{SI}$ from similar but irrelevant categories. Since we focus on the concepts that the text-to-image model can hardly recognize and visualize, $D_{TG}$ contains lots of plausible, disrupted, or even irrelevant samples, which we assign a zero score. This helps the aesthetic model understand that originally generated samples are low-quality samples. In contrast, $D_{SI}$ somehow describes some attributes for the concept but still does not match the concept, to which we assign the average score of 5.0. We employ the normal regression loss formulated as follows:

$$\mathcal{L}_{aes} = \frac{1}{|\mathbf{X}_T|} \sum_{x_t \in \mathbf{X}_T} \|p_t, \hat{p}_t\| + \frac{\lambda}{|D_{TG}| + |D_{SI}|} \sum_{x'_t \in D_{TG} \cup D_{SI}} \|p'_t - \hat{p}'_t\|, \tag{18}$$

where $p$ denotes score predicted by the original scorer, when $\hat{p}$ denotes the score we assign, and $\lambda$ is set to 1.0 in our experiments. Following this paradigm, we train an aesthetic scorer for each category of concepts, and the score personalized for each category from $D_{pool}$ can be predicted accordingly.

After selecting top-K data according to this score based on adaptor quality, we assign a lowered aesthetic score to them:

$$\hat{p}_x = 10.0 - \frac{t_{current}}{total}(10.0 - p_x), \tag{19}$$

where $p_x$ denotes the score given by a generic aesthetic scorer, $_{total}$ represents the number of total active learning cycles, when $t_{current}$ indicates the current cycle. Note that the newly selected images typically have lower quality than those from earlier cycles, we assign progressively lowered scores to them as the learning cycle proceeds.

## B.4 Adaptor Evaluation and Schedules

Following the content in the second part "The Active Schedule for Training Adaptors" in Sec. 3.3, we describe our schedule in more detail. In training CATOD, we use this indicator as a signaling proxy throughout the training process. As the number of training samples increases, the quality of selected samples at later cycles might be lower, and a fixed schedule may introduce some unnecessary details within these images (such as watermarks, borders, disturbing objects, etc.). However, these subsequent samples with relatively lower quality do contain some good photographic attributes that can help the embedding evolve. Actually, the potential problems can be alleviated by carefully-designed schedules. An intuitive illustration for this schedule is given in Fig. 6.

We first train the adaptor for 5 epochs, and save an adaptor version $A_i$ per 5 epochs, together with its corresponding generated images, constituting a sub-cycle. When this sub-cycle is completed, we filter out the adaptor with the highest quality based on the score $\gamma(A)$ (Eq.(6)). If the selected adaptor has the most training epochs across all versions, we keep the learning rate and conduct the next sub-cycle, otherwise, we will reduce the learning rate. Note that if the adaptor quality of this cycle is not as good as that of the previous cycle (because of the quality degradation caused by introducing some unnecessary details), the initial learning rate will be reduced, which helps avoid

Table 5: The statistics of the dataset we use. The test data size $\#\mathcal{D}_{val}$ of all the concepts is fixed to 100 for a fair comparison, while the training data size $\#\mathcal{D}_{pool}$ varies since different concepts since the number of data samples across publicly available datasets is different.

| Insect | CMMD ($\downarrow$) | Lizard | CMMD ($\downarrow$) | Penguin | CMMD ($\downarrow$) | Seafish | CMMD ($\downarrow$) | Snake | CMMD ($\downarrow$) |
|---|---|---|---|---|---|---|---|---|---|
| Antlion | 3.95 | Axolotl | 4.24 | Emperor Penguin Chick | 4.14 | Crampfish | 4.19 | Ahaetulla nasutar | 4.36 |
| Lycorma Delicatula | 3.99 | Frilled Lizard | 3.42 | Gentoo Penguin | 3.94 | Dragonfish | 4.16 | Aipysurus laevis | 3.56 |
| Parasitic Wasps | 4.61 | Mediterranean House Gecko | 4.45 | King Penguin Chick | 3.68 | Garfish | 3.97 | Indian Cobra | 3.91 |
| Xylotrechus | 4.19 | Oedura | 4.6 | Rockhopper Penguin | 4.04 | Tigerfish | 4.17 | Pelamis Platurus | 3.94 |
| Zopherinae | 4.12 | Opluridae | 3.88 | Royal Penguin | 3.98 | Tuna | 4.05 | Sidewinder | 4.11 |
| Thrips | 2.05 | Whiptail | 0.88 | Emperor Penguin | 1.19 | Goldfish | 1.35 | Thunder snake | 1.46 |
| Flea Beetle | 1.28 | Alligator Lizard | 1.44 | King Penguin | 0.75 | Hammerhead | 1.36 | Garter snake | 2.35 |
| Aphids | 1.16 | Gila Monster | 1.64 | Little penguin | 1.46 | Tench | 1.25 | Night Snake | 1.06 |
| Red Spider | 1.85 | Agama | 2.69 | Magellanic Penguin | 1.60 | Tiger Shark | 0.90 | Rock Python | 1.76 |
| Meadow Moth | 1.47 | Komodo Dragon | 2.00 | Adelie Penguin | 0.77 | Killer Whale | 0.70 | Hognose Snake | 1.94 |

Table 6: The statistics of the dataset we use. The test data size $\#\mathcal{D}_{val}$ of all the concepts is fixed to 100 for a fair comparison, while the training data size $\#\mathcal{D}_{pool}$ varies since different concepts since the number of data samples across publicly available datasets is different.

| Category | Concept | $\#\mathcal{D}_{pool}$ | $\#\mathcal{D}_{val}$ |
|---|---|---|---|
| Insect | Zopherinae | 1357 | 100 |
| | Antlion | 864 | 100 |
| | Lycorma Delicatula | 5108 | 100 |
| | Parasitic Wasps | 877 | 100 |
| | Xylotrechus | 1043 | 100 |
| Lizard | Axolotl | 1200 | 100 |
| | Frilled Lizard | 1008 | 100 |
| | Mediterranean House Gecko | 889 | 100 |
| | Oedura | 798 | 100 |
| | Opluridae | 818 | 100 |
| Penguin | Emperor Penguin Chick | 987 | 100 |
| | Gentoo Penguin | 3992 | 100 |
| | King Penguin Chick | 1471 | 100 |
| | Rock Hopper Penguin | 877 | 100 |
| | Royal Penguin | 754 | 100 |
| Sea Fish | Crampfish | 1179 | 100 |
| | Dragonfish | 835 | 100 |
| | Garfish | 1200 | 100 |
| | Tigerfish | 538 | 100 |
| | Tuna | 1236 | 100 |
| Snake | Ahaetulla Nasutar | 893 | 100 |
| | Aipysurus Laevis | 1015 | 100 |
| | Indian Cobra | 1200 | 100 |
| | Pelamis Platurus | 1091 | 100 |
| | Sidewinder | 1200 | 100 |

introducing disturbing elements led by some samples with insufficient quality. The sub-cycles will be continuously conducted until the minimum learning rate is reached or the adaptor quality no longer increases. Our learning rates include $5 \times 10^{-4}, 2.5 \times 10^{-4}, 7.5 \times 10^{-5}, 5 \times 10^{-5}, 2.5 \times 10^{-5}$.

## C Experiments

### C.1 How to Automatically Locate OOD Concepts

Since the publicly available dataset, i.e., ImageNet, iNaturalist 2018 [67], IP102 [71], which we adopt in our research contains 9,244 concepts with 14,883,455 images in total, it is infeasible to manually test and decide what concept is the out-of-distribution concept one by one. Therefore, our procedure for figuring out is two-fold: (1) automatically quantifying the distribution drift of the given concept from what the text-to-image model (Stable Diffusion 2.0, a.k.a SD 2.0) learned with

the CMMD metric (thanks to the recent work proposed by Google [27]); (2) Manually verifying whether the text-to-image model could generate the right content, according to the rank given by the CMMD metric above. Specifically, in the first step, we randomly sample 10 image data for each concept within the datasets and then generate 10 images with the model with the text prompt "A photo of $S^*$", in which $S^*$ denotes the concept name. Then, we calculate the CMMD discrepancy between these two data batches and record them. The CMMD scores of some concepts (including both Out-of-Distribution concepts in the first 5 rows and In-Distribution in the last 5 rows) are listed in Table 5.

Empirically, we observed that SD 2.0 is unable to synthesize the concepts with a CMMD metric above 3.5. Therefore, we pick 25 OOD concepts accordingly divide them into 5 categories, and make our dataset as follows:

- **Insect**: Zopherinae, Antlion, Lycorma Delicatula, Parasitic Wasps, Xylotrechus;
- **Lizard**: Axolotl, Frilled Lizard, Mediterranean House Gecko, Oedura, Opluridae;
- **Penguin**: Emperor Penguin Chick, Gentoo Penguin, King Penguin Chick, Rock Hopper Penguin, Royal Penguin;
- **Sea Fish**: Crampfish, Dragonfish, Garfish, Tigerfish, Tuna;
- **Snake**: Ahaetulla Nasutar, Aipysurus Laevis, Indian Cobra, Pelamis Platurus, Sidewinder.

Actually, the dataset we collect is not class-balanced, so the number of samples we can collect varies across different concepts. The statistics of each concept are shown in Table 6. For the concepts with a $\mathcal{D}_{pool}$ less than 1,000 images, we also collect some of them from image host websites including Dreamstime, Flicker, istockphoto, pinterest, and shutterstock. The dataset will be released as soon as the code is released.

## C.2    Experimental Settings

To verify the effectiveness and extensibility of our proposed CATOD framework, we conduct experiments on 25 different OOD concepts that the large-scale text-to-image (i.e. Stable Diffusion 2.0 [45]) implement different versions of CATOD. This section explains some important implementation details for the dataset, the aesthetic scorer, the concept-matching scorer we use, and how we design the learning paradigm.

**The Learning Rate Schedule.**    As shown in Algorithm 1, we have a learning rate group R, which includes 5 learning rates: $5 \times 10^{-4}$, $2.5 \times 10^{-4}$, $7.5 \times 10^{-5}$, $5 \times 10^{-5}$, $2.5 \times 10^{-5}$. Following the indicator $\gamma(A)$ in Eq. (6), we calculate this value after every epoch. When this indicator falls below the previous evaluation, we reduce the learning rate. If the learning rate cannot be lowered further, we conclude that the model has converged and the training is complete. Note that we choose to train 20 epochs in all experiments as claimed in the main context, it is possible that the training is early stopped before reaching epoch 20.

**Aesthetic Scorers.**    To offer the basic knowledge for aesthetic evaluation, we use pre-trained generic models, including ReLIC [81], TANet [23], SD-Chad Scorer, and VLAD [59]. Note that the scores given by the generic aesthetic scoring model tend to lie in the majority score range (5 to 6) in our dataset, we ought to personalize this model accordingly to our training set, which we refer to as PIAA [84]. Since PIAA is a typical small sample problem, we adopt similar experimental settings and evaluation criteria by referring to Few-Shot Learning [16] and a previous PIAA research work: PA-IAA [33].

**Concept-Matching Scorers.**    Since the first step of general image retrieval is feature extraction, we use the CLIP ViT-L/14 encoder, which is a commonly used and reliable feature extractor [2, 82, 34].

## C.3    About Compositional Results over Multiple-Concepts.

To further validate the effectiveness of CATOD, we also present compositional results based on the LoRA adaptors we obtained, as illustrated in Figure 7. We compose different types of concepts, including (i) background concepts such as brick wall and grassland for Emperor Penguin Chicks,

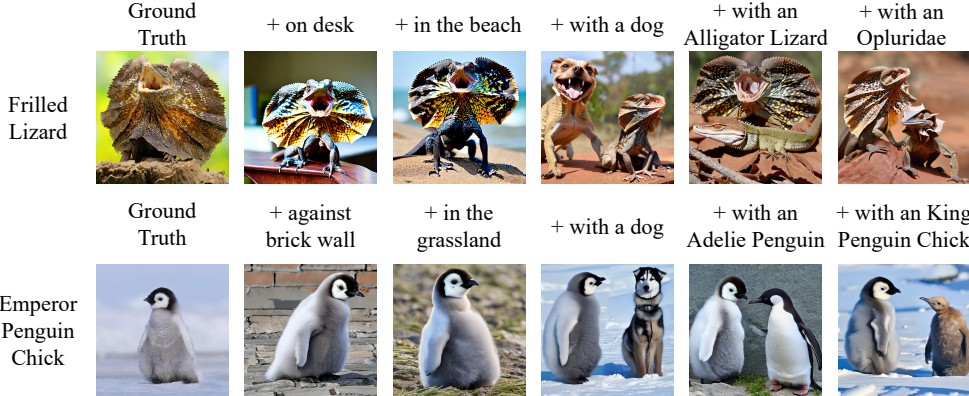

Figure 7: **Generative results with 2 concepts within one image**. Experiments are conducted based on the LoRA adaptor fully trained on concepts "Frilled Lizard" and "Emperor Penguin Chick". We try to compose these creatures with background elements, in-distribution concepts, and out-of-distribution concepts learned by other adaptors. The final results show high quality with minimal disruptive details.

and desk and beach for Frilled Lizard; (ii) in-distribution and common concepts, like the dog; (iii) in-distribution and similar concepts, like Adelie penguin for Emperor Penguin Chicks, and Alligator Lizard for Frilled Lizard; (iii) out-of-distribution concepts from other adaptors, like King Penguin Chicks for Emperor Penguin Chicks and Opluridae for Frilled Lizard. Our observations reveal that the synthetic results on out-of-distribution concepts with CATOD can seamlessly integrate different background elements, even if they were not present in real-world images (columns 1 to 2). For other in-distribution concepts like dogs, LDMs tend to represent a specific species with similar color and texture, while different concepts remain highly distinguishable within one image (columns 3 to 4). For out-of-distribution concepts, we also note that the creatures are depicted correctly, but may exhibit some confused visual details that negatively impact the aesthetics of the image (column 5).

## C.4  More Analysis.

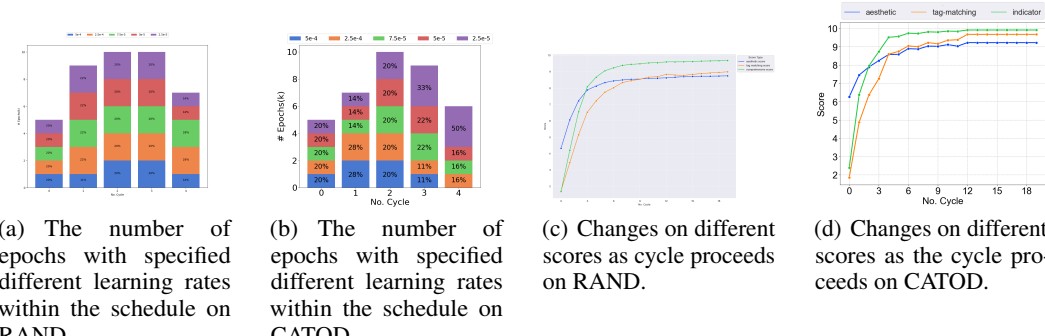

(a) The number of epochs with specified different learning rates within the schedule on RAND.

(b) The number of epochs with specified different learning rates within the schedule on CATOD.

(c) Changes on different scores as cycle proceeds on RAND.

(d) Changes on different scores as the cycle proceeds on CATOD.

Figure 8: **A comparison on how the schedule and scores change on RAND(scheduled) and CATOD as cycle proceeds on concept emperor penguin(chick).** (a),(b) show how the #epochs for each learning rate in the schedule change as the cycle proceeds, when (c),(d) show how aesthetic/concept-matching/comprehensive score change on RAND (scheduled) and CATOD. The scores for CATOD stop changing at cycle 12 since more added samples do not help boost adaptor quality.

**CATOD is more reliable on schedules.**    Figure 8(a),(b) show a comparison of how our schedules adjust as the number of cycles increases on RAND(scheduled) and CATOD. Both of them show the tendency that training epochs with larger learning rates decrease when those with small learning rates

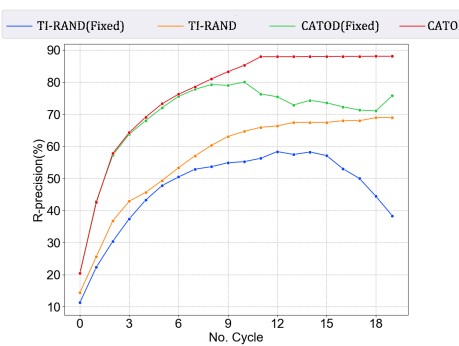

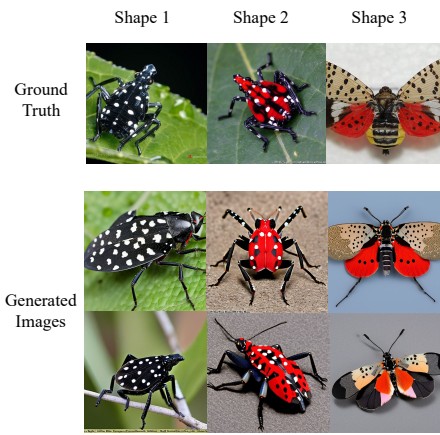

(a) **R-precision(%) with different strategies as training pool expands.** The experiments are performed over concept "emperor penguin(chick)" and produced over RAND and CATOD. To further show the impact of the number of samples, we also compare their modified versions that do not use dynamic schedules.

(b) **A comparison on real-images with CATOD-generated ones in 3 different shapes of lycorma delicatula.** We can see that with a carefully designed selection, the adaptor helps produce all the shapes of the concept "lycorma delicatula".

Table 7: A Comparison over the performance of CATOD on different types of the initial training data pool, in terms of the CLIP score and CMMD score with 100 images sampled at last. This table shows the average result of 5 sub-classes within each category. The overall improvement of our proposed CATOD is provided by "Imp.". Methods with the best performance are bold-folded.

| Comparison Methods | CLIP Score (↑) | | | | | | CMMD [27] (↓) | | | | | |
|---|---|---|---|---|---|---|---|---|---|---|---|---|
| | insect | lizard | penguin | seafish | snake | Avg. | insect | lizard | penguin | seafish | snake | Avg. |
| 10 HQ initial samples | 73.87 | 79.23 | 75.59 | 83.84 | 82.19 | 78.94 | 0.87 | 0.80 | 0.71 | 0.79 | 0.78 | 0.79 |
| 10 RAND initial samples | 72.26 | 74.06 | 72.89 | 79.75 | 78.08 | 75.41 | 0.92 | 0.84 | 0.73 | 0.80 | 0.93 | 0.84 |
| 20 HQ initial samples | 75.18 | 77.59 | 76.97 | 82.78 | 83.14 | 79.13 | 0.85 | 0.79 | 0.69 | 0.72 | 0.75 | 0.76 |
| 20 RAND initial samples | 72.08 | 72.82 | 72.97 | 77.74 | 75.06 | 74.13 | 1.01 | 1.03 | 0.92 | 0.97 | 0.83 | 0.95 |
| 50 HQ initial samples | 75.27 | 77.27 | 76.85 | 82.90 | 83.37 | 79.13 | 0.84 | 0.80 | 0.63 | 0.69 | 0.71 | 0.73 |
| 50 RAND initial sample | 70.34 | 70.53 | 67.49 | 72.10 | 73.74 | 70.84 | 1.13 | 1.29 | 0.94 | 1.06 | 0.97 | 1.08 |

increase, meaning that the training schedule focuses more on fine-tuning at later cycles. Comparing CATOD with RAND, we can see that the epochs with the maximal learning rate already diminish at cycle 4 on CATOD while RAND still needs large learning rates, indicating that CATOD are better at recognizing the given concept and tends to be more stable.

**CATOD achieves a good score earlier than other methods.** Figure 8(c),(d) shows a comparison of how the aesthetic/concept-matching/comprehensive scores change as the cycle proceeds on RAND and CATOD. We can see that CATOD already achieves a higher value on all scores and cannot be further boosted with 120 training samples on cycle 12 when the performance for RAND still fluctuates. This phenomenon illustrates that our active selection strategy makes our embedding be trained on more high-quality data.

Furthermore, we also notice that aesthetic/concept-matching scores fluctuate at later cycles when the comprehensive score is enhanced continuously, which exactly corresponds to our proposed dual scoring system that balances the importance of these two factors when ensuring the quality of the adaptor never declines.

**The performance of CATOD on different initial training data pool.** We also ablate CATOD over the initial training set size/quality. In detail, we retest CATOD with different numbers of initial samples (10,20,50) and different quality, i.e., high-quality (HQ) and random sampling (RAND), with 100 images sampled at last and 10 images selected per cycle. These experiments are tested with adaptor LoRA. The results are shown in Table 7:

Table 8: A Comparison over the performance of CATOD on different architectures, in terms of the CLIP score and CMMD score with 100 images sampled at last. This table shows the average result of 5 sub-classes within each category. The overall improvement of our proposed CATOD is provided by "Imp.". Methods with the best performance are bold-folded.

| Comparison Methods | CLIP Score (↑) | | | | | | | CMMD [27] (↓) | | | | | | |
| --- | insect | lizard | penguin | seafish | snake | Avg. | Imp. | insect | lizard | penguin | seafish | snake | Avg. | Imp. |
| SD 1.5 + RAND | 63.70 | 64.94 | 65.98 | 66.29 | 69.43 | 66.07 | ⇑9.27 | 1.49 | 1.47 | 1.65 | 1.62 | 1.33 | 1.51 | ⇓0.61 |
| SD 1.5 + CLIP | 68.78 | 70.38 | 70.89 | 74.20 | 73.89 | 71.63 | ⇑3.71 | 1.19 | 1.45 | 1.28 | 1.45 | 1.21 | 1.32 | ⇓0.42 |
| SD 1.5 + CATOD | **71.73** | **74.79** | **73.45** | **79.37** | **77.35** | **75.34** | - | **1.01** | **0.97** | **0.78** | **0.92** | **0.84** | **0.90** | - |
| SDXL + RAND | 72.39 | 73.04 | 71.75 | 74.81 | 70.47 | 72.49 | ⇑8.05 | 1.39 | 1.45 | 1.49 | 1.58 | 1.01 | 1.38 | ⇓0.50 |
| SDXL + CLIP | 79.32 | 78.24 | 74.98 | 82.16 | 79.75 | 78.89 | ⇑1.65 | 1.16 | 1.27 | 1.09 | 0.93 | 1.07 | 1.10 | ⇓0.22 |
| SDXL + CATOD | **80.37** | **79.58** | **77.56** | **85.20** | **79.97** | **80.54** | - | **0.95** | **0.87** | **0.96** | **0.89** | **0.75** | **0.88** | - |

Table 9: A Comparison over the diversity score of CATOD, in terms of the CLIP score and CMMD score with 100 images sampled at last. This table shows the average result of 5 sub-classes within each category. The overall improvement of our proposed CATOD is provided by "Imp.". Methods with the best performance are bold-folded.

| Comparison Methods | LPIPS Score(↑) | | | | | | |
| --- | insect | lizard | penguin | seafish | snake | Avg. | Imp. |
| DreamBooth + CLIP | 0.254 | 0.156 | 0.149 | 0.305 | 0.265 | 0.226 | ⇑0.052 |
| DreamBooth + CATOD | 0.362 | 0.198 | 0.197 | 0.349 | 0.286 | 0.278 | - |
| TI + CLIP | 0.198 | 0.257 | 0.174 | 0.108 | 0.208 | 0.189 | ⇑0.033 |
| TI + CATOD | 0.267 | 0.278 | 0.152 | 0.278 | 0.133 | 0.222 | - |
| LoRA + CLIP | 0.178 | 0.184 | 0.155 | 0.203 | 0.094 | 0.163 | ⇑0.054 |
| LoRA + CATOD | 0.245 | 0.203 | 0.217 | 0.244 | 0.178 | 0.217 | - |

From these results, we can draw the following conclusions:

- Initial batch size has a more significant impact on randomly initialized samples than on high-quality samples. Specifically, the performance change with high-quality samples is at most 1.40 in the CLIP score and 0.10 in the CMMD score concerning different initial numbers of training data. In contrast, the performance change on low-quality is up to 7.65 in the CLIP score and 0.45 in the CMMD score.

- The quality of initial samples does have an impact on generative results since we can see a consistent performance loss when changing HQ initial samples to randomly initialized samples. With the initial size increase, this impact tends to be even more significant.

## C.5 Further Extension: Concept with multiple shapes

Recall that Stable Diffusion 2.0 fails to generate emperor penguin chicks, which accounts for the fact that adult emperor penguin chicks and their chicks have different appearances. We raise another question: can we train an adaptor with a concept that has different shapes? Inspired by this, we also test our proposed CATOD on the concept "lycorma delicatula", which is a kind of pest with 3 different forms in its life-cycle, and surprisingly find that with 300 training samples, the best version of adaptors successfully produces all these 3 shapes, as shown in Figure 9(b). These results further validate the effectiveness of our framework, even in the presence of appearance ambiguity.

## C.6 About Experimental Results on other Architectures

Our experiments in the main content are all conducted on Stable Diffusion 2.0 (SD 2.0). To further validate the superiority of our proposed CATOD, we conduct additional experiments on other architectures (based on LoRA), including Stable Diffusion 1.5 (SD 1.5) and Stable Diffusion XL (SDXL). The results are shown in Table 9.

This table shows that our proposed CATOD is also compatible with different architectures with notable performance gain compared to the baselines.

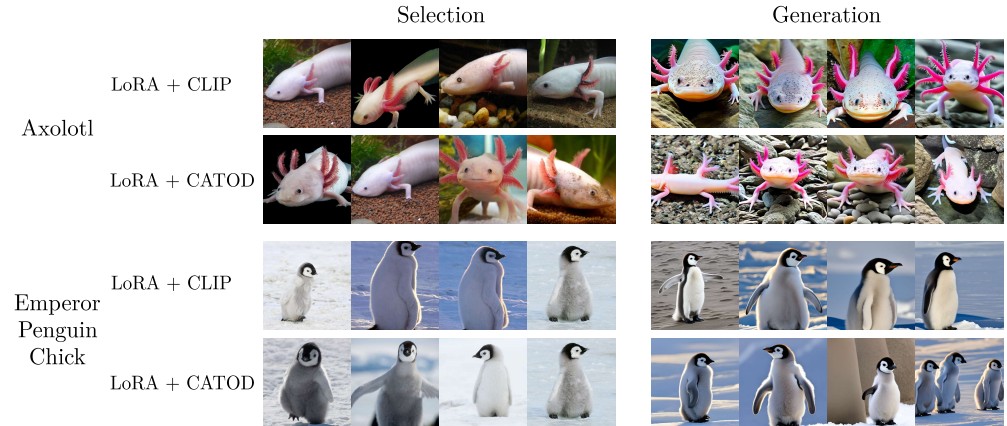

Figure 9: A comparison of selected and generate samples on different combinations of methods and concepts. We can observe that training samples with different angles selected by CATOD also lead to diverse angle in their generative results.

## C.7 About the diversity

CATOD maintains the diversity to produce OOD concepts with different angles or poses in generative results. To validate the diversity of our generative results, we first provide a quantitative evaluation with LPIPS [80, 78]. The results are shown in Table 9. In this table, we can see that CATOD gives out a diversity improvement up to 0.17 in the LPIPS score compared to CLIP-based sampling, and outperforms CLIP over most categories. From this, we conclude that CATOD also preserves diversity. Since the training set contains samples with different angles in CATOD, it is also easy to produce objects with different angles, and we show the examples on concepts "Axolotl" and "Emperor Penguin Chick" in Figure 9. We can observe that training samples with different angles selected by CATOD also lead to diverse angle in their generative results.

