# OpenReview forum: "Locating What You Need: Towards Adapting Diffusion Models to OOD Concepts In-the-Wild"
_NeurIPS.cc/2024/Conference — NeurIPS 2024 poster_

### Official Review · Reviewer_uVj3 · 2024-07-07

**Soundness:** 3
**Presentation:** 3
**Contribution:** 3
**Rating:** 7
**Confidence:** 4

**Summary:**

The paper introduces CATOD, a framework designed to enhance the adaptation of text-to-image models for out-of-distribution (OOD) concepts. It addresses the issue of low-quality training data by employing an active learning approach that iteratively improves the training set. The framework utilizes a scoring system comprising aesthetic and concept-matching scores to guide data selection and training. This scoring system dynamically determines the relative weight of the aesthetic score and content matching score to prioritize samples for selecting the training set. The authors demonstrate significant performance improvements using CATOD, achieving notable improvements on standard metrics.

**Strengths:**

1. The paper is well-written and easy to follow.
2. The research problem of out-of-distribution (OOD) on text-to-image models is essential, and the motivation is quite clear.
3. The approach is innovative, which combines active learning with a novel scoring system to enhance model adaptation.
4. The paper has very adequate quantitative experimental results and show clear improvements over baseline methods.

**Weaknesses:**

1. The paper does not provide enough detail to clarify how the CMMD metric differs from other commonly-used metrics for image generation, such as Inception Score and FID. The authors should further explain this.
2. Since your proposed CATOD is trained on few samples (100 for single-concept and 200 for multi-concept), the generation model may be biased toward specific samples. Reporting diversity metrics of the generated samples can help clarify this issue.

**Questions:**

1. I wonder if your proposed CATOD also outperforms other methods on previous metrics such as FID and the Inception Score.
2. In the paper, random sampling and CLIP-score-based sampling are used as the comparison method, and finally 100-200 images are selected for training. Would it be better to fine-tune the model directly using all 1000 images?
3. In addition to the three methods mentioned in the paper, are there other custom content generation methods that your proposed CATOD framework is still applicable to, such as LyCORIS[1] and ELITE[2]?

[1] YEH, SHIH-YING, Yu-Guan Hsieh, Zhidong Gao, Bernard BW Yang, Giyeong Oh, and Yanmin Gong. "Navigating text-to-image customization: From lycoris fine-tuning to model evaluation." In *The Twelfth International Conference on Learning Representations*. 2023.

[2] Wei, Yuxiang, Yabo Zhang, Zhilong Ji, Jinfeng Bai, Lei Zhang, and Wangmeng Zuo. "Elite: Encoding visual concepts into textual embeddings for customized text-to-image generation." In *Proceedings of the IEEE/CVF International Conference on Computer Vision*, pp. 15943-15953. 2023.

**Limitations:**

The authors have adequately addressed the limitations.

---

> ### Author Rebuttal · Authors · 2024-08-05
>
> Thank for the good words! We are happy that you enjoyed the paper!
>
> **Part 1: Why CMMD is a good metric (W1, Q1)**
>
> As reported in many recent works [1,2,3], the most popular image-matching metrics like Inception Score, Precision/Recall, and FID (Frechet Inception Distance) may disagree with human raters, thus ill-suited for evaluating recent text-to-image models. In comparison, CMMD uses CLIP embeddings and Maximum Mean Discrepancy (MMD) that correctly ranks the image sets based on the severity of the distortion. The limitations of FID and previous metrics can be listed as follows:
>
> |                      | FID and other previous metrics                               | MMD distance                                                 |
> | -------------------- | ------------------------------------------------------------ | ------------------------------------------------------------ |
> | Inception embeddings | × Weak image embeddings<br />× Normality assumption<br />× Sample inefficient<br />× Biased estimator | × Weak image embeddings<br />√  Distribution-free<br />√  Sample inefficient<br />√  Unbiased estimator |
> | CLIP embeddings      | √ Rich image embeddings<br />× Normality assumption<br />× Sample inefficient<br />× Biased estimator | √  Rich image embeddings<br />√  Distribution-free<br />√  Sample inefficient<br />√  Unbiased estimator |
>
> Also, we are glad to report our experimental results with FID, Precision, and Recall as follows based on CATOD adapted through LoRA:
>
> | Insect       | FID   | Precision | Recall |
> | ------------ | ----- | --------- | ------ |
> | LoRA + RAND  | 35.02 | 0.55      | 0.83   |
> | LoRA + CLIP  | 28.46 | 0.85      | 0.91   |
> | LoRA + CATOD | 26.74 | 0.92      | 0.94   |
>
> | Penguin      | FID   | Precision | Recall |
> | ------------ | ----- | --------- | ------ |
> | LoRA + RAND  | 11.17 | 0.63      | 0.79   |
> | LoRA + CLIP  | 5.22  | 0.80      | 0.87   |
> | LoRA + CATOD | 4.03  | 0.84      | 0.88   |
>
> [1] Jayasumana, Sadeep, Srikumar Ramalingam, Andreas Veit, Daniel Glasner, Ayan Chakrabarti, and Sanjiv Kumar. "Rethinking fid: Towards a better evaluation metric for image generation." *arXiv preprint arXiv:2401.09603* (2023).
>
> [2] Grimal, Paul, Hervé Le Borgne, Olivier Ferret, and Julien Tourille. "TIAM-A Metric for Evaluating Alignment in Text-to-Image Generation." In *Proceedings of the IEEE/CVF Winter Conference on Applications of Computer Vision*, pp. 2890-2899. 2024.
>
> [3] Lee, Tony, Michihiro Yasunaga, Chenlin Meng, Yifan Mai, Joon Sung Park, Agrim Gupta, Yunzhi Zhang, et al. "Holistic evaluation of text-to-image models." *Advances in Neural Information Processing Systems* 36 (2024).
>
> **Part 2: About the diversity evaluation (W2)**
>
> CATOD maintains the diversity to produce OOD concepts with different angles or poses in generative results. To validate the diversity of our generative results, we further provide a quantitative evaluation with LPIPS~[1,2]. The results are shown in the Table below:
>
> | Comparison Methods | Insect    | Lizard    | Penguin   | Seafish   | Snake     |
> | ------------------ | --------- | --------- | --------- | --------- | --------- |
> | DreamBooth + CLIP  | 0.254     | 0.156     | 0.149     | 0.305     | 0.265     |
> | DreamBooth + CATOD | **0.362** | **0.198** | **0.197** | **0.349** | **0.286** |
> | TI + CLIP          | 0.198     | 0.257     | **0.174** | 0.108     | **0.208** |
> | TI + CATOD         | **0.267** | **0.278** | 0.152     | **0.278** | 0.133     |
> | LoRA + CLIP        | 0.178     | 0.184     | 0.155     | 0.203     | 0.094     |
> | LoRA + CATOD       | **0.245** | **0.203** | **0.217** | **0.244** | **0.178** |
>
> In this table, we can see that CATOD gives out a diversity improvement up to 0.17 in the LPIPS score compared to CLIP-based sampling, and outperforms CLIP over most categories. From this, we conclude that CATOD also preserves diversity. Since the training set contains samples with different angles, it is also easy to produce objects with different angles. We will surely add a visualization of our selected samples and generative results with different poses/angles in our final revision.

---

> ### Author Response · Authors · 2024-08-05
>
> **Part 3: Using 1000 data is not feasible (Q2)**
>
> Using all the 1000 data for training is not feasible for adapting OOD concepts. The experimental results of using all data are as follows:
>
> | Combinations          | Axolotl (CLIP$\uparrow$) | Axolotl (CMMD$\downarrow$) | Emperor Penguin Chick (CLIP$\uparrow$) | Emperor Penguin Chick (CMMD$\downarrow$) |
> | --------------------- | ------------------------ | -------------------------- | -------------------------------------- | ---------------------------------------- |
> | DreamBooth + RAND     | 66.19                    | 1.13                       | 67.34                                  | 1.54                                     |
> | DreamBooth + ALL DATA | 68.24                    | 1.01                       | 70.12                                  | 1.23                                     |
> | DreamBooth + CATOD    | 72.25                    | 0.88                       | 74.38                                  | 0.83                                     |
> | TI + RAND             | 59.67                    | 2.11                       | 62.89                                  | 2.10                                     |
> | TI + ALL DATA         | 52.79                    | 3.04                       | 55.63                                  | 2.67                                     |
> | TI + CATOD            | 69.95                    | 1.49                       | 68.14                                  | 1.15                                     |
> | LoRA + RAND           | 65.35                    | 1.41                       | 68.23                                  | 1.58                                     |
> | LoRA + ALL DATA       | 67.24                    | 1.47                       | 71.17                                  | 1.33                                     |
> | LoRA + CATOD          | 72.87                    | 1.37                       | 73.19                                  | 1.13                                     |
>
> As we see in this table, performing adaptation over the full dataset can lead to severe performance loss compared to using carefully selected ones. On Textual Inversion (TI), it even becomes worse than random sampling. This is due to that the full dataset contains lots of "bad" samples, which exhibit too many disruptive elements -- in turn -- introducing wrong details that mislead the concept adaptation.
>
> **Part 4: Generalizing CATOD to other customization methods. (Q3)**
>
> Thanks for the advice! CATOD can be easily generalized to ELITE and LyCORIS since it is designed as a general framework for the adapters. We show the results for applying CATOD to ELITE/LyCORIS on concepts "axolotl" and "emperor penguin chick" as follows:
>
> | Combinations    | Axolotl (CLIP) | Axolotl (CMMD) | Emperor Penguin Chick (CLIP) | Emperor Penguin Chick (CMMD) |
> | --------------- | -------------- | -------------- | ---------------------------- | ---------------------------- |
> | LoRA + CLIP     | 72.87          | 1.37           | 73.19                        | 1.13                         |
> | ELITE + CLIP    | 74.14          | 1.03           | 74.37                        | 1.01                         |
> | LyCORIS + CLIP  | 72.95          | 1.18           | 74.08                        | 1.08                         |
> | LoRA + CATOD    | 75.35          | 0.85           | 74.89                        | 0.79                         |
> | ELITE + CATOD   | 75.68          | 0.83           | 74.68                        | 0.74                         |
> | LyCORIS + CATOD | 76.04          | 0.79           | 75.05                        | 0.73                         |
>
> From these results, we can also see that CATOD gives a consistent boost to ELITE and LyCORIS compared to other sampling strategies. Despite that CATOD is compatible with ELITE and LyCORIS, they do not exhibit a notable performance gain compared to LoRA.

---

### Official Review · Reviewer_bHo6 · 2024-07-08

**Soundness:** 3
**Presentation:** 3
**Contribution:** 3
**Rating:** 6
**Confidence:** 5

**Summary:**

This paper introduces a novel framework called Controllable Adaptor Towards Out-of-Distribution (OOD) Concepts (CATOD). CATOD is designed to adapt text-to-image models to OOD concepts and generate high-quality images of those OOD concepts accordingly. The authors identified the challenge of accurately depicting OOD concepts due to low-quality training data. CATOD employs an active learning approach to iteratively accumulate high-quality training data and update the adaptor. The framework incorporates an aesthetic score and a concept-matching score to guide the accumulation of training data. Extensive experiments demonstrate significant improvements in both the CLIP score and the CMMD metric.

**Strengths:**

1. The presentation of the paper is clear and concise. The method is easy to understand and the delivery of the paper is smooth.
2. The paper addresses a significant challenge in text-to-image generation by focusing on OOD concepts, and serves as the first work to address the OOD challenge from a data-centric perspective.
3. The method is well-motivated and supported by theoretical insights into the importance of aesthetics and concept-matching scores.
4. The experimental results demonstrate significant improvements over existing methods.

**Weaknesses:**

1. To support the claim in Line 189 that your selection "guarantees the sample diversity", the authors should report the diversity metrics of the generated samples.
2. The paper provides ablation results on the design of weighted scoring strategies, which is valuable. I think more extensive ablation studies on the impact of initial data size, the type of text-to-image encoders, etc. can help signify the paper's contribution.
3. To further investigate how the weighting strategy work in your proposed CATOD, it would better to list how the weights of aesthetic/concept-matching scores change as your learning cycle proceeds over different concepts.

**Questions:**

1. Following weakness 1, can you compare the diversity of the fine-tuned model under different strategies?
2. It would be intriguing to investigate how initial data size impact the experiments, especially for the multiple concept scenario as you mentioned in Sec. 5.2 following weakness 2.
3. For different concepts, which of the two scores gets high weightage (in equation 11) as the training progresses?

**Limitations:**

See weakness.

---

> ### Author Rebuttal · Authors · 2024-08-05
>
> Thank you for your comments and suggestions! We hope that our rebuttal addresses your concerns.
>
> **Part 1: About the diversity evaluation (W1, Q1)**
>
> CATOD maintains the diversity to produce OOD concepts with different angles or poses in generative results. To validate the diversity of our generative results, we further provide a quantitative evaluation with LPIPS~[1,2]. The results are shown in the Table below:
>
> | Comparison Methods | Insect    | Lizard    | Penguin   | Seafish   | Snake     |
> | ------------------ | --------- | --------- | --------- | --------- | --------- |
> | DreamBooth + CLIP  | 0.254     | 0.156     | 0.149     | 0.305     | 0.265     |
> | DreamBooth + CATOD | **0.362** | **0.198** | **0.197** | **0.349** | **0.286** |
> | TI + CLIP          | 0.198     | 0.257     | **0.174** | 0.108     | **0.208** |
> | TI + CATOD         | **0.267** | **0.278** | 0.152     | **0.278** | 0.133     |
> | LoRA + CLIP        | 0.178     | 0.184     | 0.155     | 0.203     | 0.094     |
> | LoRA + CATOD       | **0.245** | **0.203** | **0.217** | **0.244** | **0.178** |
>
> In this table, we can see that CATOD gives out a diversity improvement up to 0.17 in the LPIPS score compared to CLIP-based sampling, and outperforms CLIP over most categories. From this, we conclude that CATOD also preserves diversity. Since the training set contains samples with different angles, it is also easy to produce objects with different angles. We will surely add a visualization of our selected samples and generative results with different poses/angles in our final revision.
>
> **Part 2: Ablating initial size/quality in CATOD (W2, Q2)**
>
> It is intriguing to ablate CATOD over the initial training set size and quality! In detail, we retest CATOD with different numbers of initial samples (10,20,50) and different quality (high-quality and random sampling), with 100 images sampled at last and 10 images selected per cycle. These experiments are tested with adaptor LoRA. The results are shown as follows:
>
> | Initial Setting         | Axolotl (CLIP) | Axolotl (CMMD) | Emperor Penguin Chick (CLIP) | Emperor Penguin Chick (CMMD) |
> | ----------------------- | -------------- | -------------- | ---------------------------- | ---------------------------- |
> | 10 HQ initial samples   | 73.99          | 0.84           | 74.97                        | 0.71                         |
> | 10 RAND initial samples | 72.89          | 0.93           | 73.95                        | 0.75                         |
> | 20 HQ initial samples   | 75.35          | 0.82           | 74.89                        | 0.69                         |
> | 20 RAND initial samples | 72.27          | 1.10           | 72.38                        | 0.73                         |
> | 50 HQ initial samples   | 75.37          | 0.81           | 74.56                        | 0.63                         |
> | 50 RAND initial samples | 70.39          | 1.34           | 68.85                        | 1.12                         |
>
> From these results, we may draw the following conclusions:
>
> 1. With HQ initial samples, the initial batch size does not have a significant compact, since a change in this size is not more than 0.3 in the CLIP score and 0.08 in the MMD score. However, this initial size has a significant impact with randomly initialized samples with at most a 5.71 loss in CLIP score and 0.49 in CMMD. We account this phenomenon for that randomly initialized images contain more bad-quality ones that mislead the adaptation.
> 2. The quality of initial samples does have an impact on generative results since we can see a consistent performance loss when changing HQ initial samples to randomly initialized samples. With the initial size increase, this impact tends to be even more significant.
>
> The full results will be added in our revisions.
>
> **Part 3: Ablating the text-to-image encoder (W2)**
>
> Thanks for the advice! It is straightforward to extend our experiments to other text-to-image encoders (like SD 1.5 and SDXL). We show some of these results with LoRA in the tables below:
>
> | T2I Model Ablation        | Axolotl (CLIP) | Axolotl  (CMMD) | Emperor Penguin Chick (CLIP) | Emperor Penguin Chick (CMMD) |
> | ------------------------- | -------------- | --------------- | ---------------------------- | ---------------------------- |
> | CATOD + SD 2.0 (original) | 75.35          | 0.85            | 74.89                        | 0.79                         |
> | CATOD + SD 1.5            | 74.79          | 0.88            | 73.01                        | 0.92                         |
> | CATOD + SDXL              | 79.36          | 0.72            | 80.37                        | 0.66                         |
>
> This table shows that our proposed CATOD is also compatible with different text-to-image encoders with notable performance. We will add these ablations to our final revisions.
>
> **Part 4: How the weight of different scores changes (W3, Q3)**
>
> Following Eq. (11) in the main context, the overall aesthetic score $\gamma_{aes}(A)$, overall concept matching score $\gamma_{con}(A)$​ and the corresponding weights on concept "emperor penguin chick" can be listed as follows:
>
> | Cycle                   | 1     | 2     | 3     | 4     | 5     |
> | ----------------------- | ----- | ----- | ----- | ----- | ----- |
> | $\gamma_{aes}(A)$       | 6.26  | 8.57  | 9.03  | 9.11  | 9.22  |
> | aesthetic weight        | 0.374 | 0.143 | 0.097 | 0.099 | 0.078 |
> | $\gamma_{con}(A)$       | 1.86  | 8.59  | 8.54  | 9.35  | 9.67  |
> | concept-matching weight | 0.814 | 0.141 | 0.146 | 0.065 | 0.033 |
>
> During the first three cycles, concept-matching takes precedence over aesthetics because the OOD concepts are not yet fully learned by the adapter. As the cycles progress, concept-matching is quickly achieved. In the last two cycles, the adapter stabilizes and focuses more on enhancing aesthetics.

---

### Official Review · Reviewer_PRf3 · 2024-07-10

**Soundness:** 3
**Presentation:** 4
**Contribution:** 4
**Rating:** 7
**Confidence:** 4

**Summary:**

This work tackles the challenge of adapting text-to-image diffusion models to out-of-distribution (OOD) concepts. The authors introduce a framework called Controllable Adaptor Towards Out-of-Distribution Concepts (CATOD), which employs an active learning paradigm to iteratively accumulate high-quality training data and optimize adaptor training. The framework features a dual scoring system that balances aesthetics and concept matching, ensuring improved quality and conciseness of the generated images.

**Strengths:**

1. This paper is easy to follow due to its clear writing, concise figures, and well-structured formulation of problem definitions, methods, and theories.
2. The motivation of this paper is innovative, addressing the overlooked problem of adapting recent text-to-image (T2I) models to out-of-distribution concepts.
3. The paper presents sufficient experimental results that effectively support its claims.

**Weaknesses:**

1.	The key contribution of this paper is Equation (6) and (7), which are claimed to be use as a signal to reduce the learning rate, stop the training in time, and select samples for the next cycle.  It seems the innovation is that the authors identify two relevant factors and utilize the strategy of active learning. Am I correct?
2.	Authors use MMSE to reinterpret the loss function of the LDM and decompose it into two terms (Eq. 10). As the description in lines 213 to 220 of the paper, associating these two terms directly with aesthetic preservation and concept-matching score is rather far-fetched and vague. Kindly advise if I am wrong.
3.	If the aesthetic and concept-matching scores are really that important, why just use those two scores to pick suitable samples for adaptation from the beginning? This would also avoid the later calculations.

**Questions:**

1. Is the CMMD shown in Figure 1 calculated using Real Images and before adapting images?
2. Are the high-quality samples shown in Figure 2 generated by vanilla Textual Inversion, DreamBooth, and LoRA? Or are they already using the CATOD framework?
3. Is “if r_k is the closet representation for r_k” in line 145 of the paper correct?
4. Eq. 6 and the description in lines 173 to 174 of the paper confuse me. Shouldn’t the maximum value of \gamma (A) be \gamma_aes (A)=\gamma_con (A)=10+40k, k=0...n, and the minimum value be \gamma_aes (A)=30+40k, or \gamma_con (A)=30+40k, k=0...n?
5. Associating the two terms of Eq. 10 directly with aesthetic and concept-matching scores requires further theoretical explanation and empirical analysis.
6. I noticed in the experiment section that the training period is about 20 epochs. Would this somehow be too less? If there were enough training cycles, we would get an overfitted LDM capable of generating images containing specific OOD concepts.
7. In Algorithm 1, lines 9 to 19, it does not look like new training samples are being selected for each epoch.
8. Does the learning rate set R determine the stopping of the training? How would you guarantee that the model has converged at this point? As question 5, a sufficiently overfitted LDM should be able to generate these OOD concepts.
9. This paper tests the proposed CATOD on SD 2.0. Have you conducted experiments to evaluate its performance on other text-to-image models, such as SD 1.5 and SDXL? Including these results would help verify the versatility of your method.
10. I am curious about how CATOD performs with different initial training set sizes (e.g., 10 and 50) and varying quality. Adding this as an ablation experiment would provide valuable insights into the robustness of your method.
11. I am also interested in whether similar categories (e.g., other kinds of penguins) might be distorted after adaptation. Have you conducted any experiments to evaluate the degree of distortion in these cases?
12. Since CMMD is a newly proposed metric by Google in CVPR 2024, it would be helpful to list the advantages of using CMMD compared to traditional metrics like Inception Score, FID, and MMD. This would help readers understand the rationale behind choosing CMMD.

---

> ### Author Rebuttal · Authors · 2024-08-05
>
> Thank you for your comments and suggestions. We have carefully addressed your concerns as follows:
>
> **Part 1: About the key technical contribution (W1)**
>
> The key technical contributions of this paper are twofold: (1) our method for estimating the impact of each sample on the model without training, corresponding to the aesthetic/concept-matching scorer described in Section 3.2; (2) our approach to effectively utilizing these scores and balancing the trade-off between the two impact factors (Eq. 6 and Eq. 7). This is achieved through Active Learning, a vital paradigm that enables dynamic trade-offs cycle-by-cycle.
>
> **Part 2: A more detailed explanation for the theories (W2, Q5)**
>
> We are pleased to provide a detailed explanation of the connection between our proposed method and the underlying theories!
>
> As introduced in our background Section (Sec.2), the LDMs consist of two core components: An encoder $\mathcal{E}$ learns to map images $x$ to a latent code $z=\mathcal{E}(x)$, and a diffusion model trained to produce code conditioned on texts. Let $c\_\theta(y)$ be a model mapping the conditioning text $y$ into latent vectors, $A$ the adapter embedded into the diffusion model, the LDM loss is given by:
> $L\_{LDM}(x,A)= \mathbb{E}\_z\sim\mathcal{E}(x),t,\epsilon\sim\mathcal{N}(0,1)\left[\Vert \epsilon-\epsilon\_{\theta, A}(z\_t,t,c\_{\theta, A}(y)) \Vert\_2\^2\right]$ in Eq.1, where t is the time step, $z\_t$ the latent noised to time $t$, $\epsilon$ the noise sample, $\epsilon\_\theta$ the denoising network. When optimizing the LDM loss, there are two important factors we should consider: the adapter $A$ that adapts the underly model to OOD image $x$, and the conditional text $y$ for the images $x$ to match. As we claimed in the main context that disrupting $x$'s can lead to misunderstanding of concept $y$, our objective is in Eq.2:
>
> $A^*,X\_{T}^*=\mbox{argmin}\_{A,X\_T} L\_{\mbox{LDM}}(x,A,y)$.
>
> Since the optimal set for $A, X_T$ is initially unknown and cannot located at once, is common to use an iterative paradigm that alternatively optimizes $A$ and $X_T$. We have the following conjugate forms:
>
> $\mathbf{X}\_T\^{(t)} = \mbox{argmin}\_{\mathbf{X}\_T\subset D\_{pool}}\mathbb{E}\_{x\sim \mathbf{X}\_T}L\_{LDM}(x,A\^{(t-1)},y)$ (Eq.3 in main context)
>
> $A\^{(t)} = \mbox{argmin}\_{A}\mathbb{E}\_{x\sim \mathbf{X}\_T\^{(t-1)}}L\_{LDM}(x,A,y)$ (Eq.4 in main context)
>
> For the first equation (Eq.3), we optimize the training $X_T$ towards both conditional text $y$ (image-text matching) and loss reduction (aesthetic preserving), which in turn leads to the motivation of decoupling aesthetic/concept-matching scores. Theorem 4.3 verifies the necessity of this decomposition while Section 3.2 implements this decomposition. The second equation (Eq.4) just performs an ordinary adaptation on select samples.
>
> Since the iterative paradigm in Eq. 3 and Eq. 4 may lead to convergence to local optima, we also need a dynamic weighted scoring system to trade-off between aesthetics and concept-matching, to alleviate the potential bias towards only one of them (leading to the practical implementation in Sec 3.3). To this end, we have explained the motivation of our theory and how we organize our CATOD.
>
> **Part 3: The pre-calculated scores are not feasible (W3)**
>
> Since Out-of-Distribution (OOD) concepts are usually unseen by both the underlying text-to-model and the aesthetic/concept-matching scorers, it is likely that the preset scorers provide inaccurate results. To address this, we must use the currently available high-quality samples to correct these inaccuracies. Following the iterative paradigm outlined in Part 2, the scorers and adaptors are alternately adjusted to ensure accurate results. Additionally, we update the scorers after selecting high-quality samples in each cycle. To demonstrate this experimentally, we conduct another ablation experiment with pre-calculated scores, as shown in the following table:
>
> | Method                           | Lizard (CLIP$\uparrow$) | penguin(CLIP$\uparrow$) | Lizard (CMMD$\downarrow$) | penguin(CMMD$\downarrow$) |
> | -------------------------------- | ----------------------- | ----------------------- | ------------------------- | ------------------------- |
> | CATOD                            | 77.00                   | 74.11                   | 0.89                      | 0.71                      |
> | CATOD with pre-calculated scores | 75.29                   | 73.57                   | 0.93                      | 0.76                      |
>
> We observe a noticeable performance loss when the scores are pre-calculated.
>
> **Part 4: Other architectures (Q9).**
>
> Thanks for the advice! It is straightforward to extend our experiments to other text-to-image models (like SD 1.5 and SDXL). We show some of these results with LoRA in the tables below:
>
> | T2I Model Ablation        | Axolotl (CLIP) | Axolotl  (CMMD) | Emperor Penguin Chick (CLIP) | Emperor Penguin Chick (CMMD) |
> | ------------------------- | -------------- | --------------- | ---------------------------- | ---------------------------- |
> | CATOD + SD 2.0 (original) | 75.35          | 0.85            | 74.89                        | 0.79                         |
> | CATOD + SD 1.5            | 74.79          | 0.88            | 73.01                        | 0.92                         |
> | CATOD + SDXL              | 79.36          | 0.72            | 80.37                        | 0.66                         |
>
> This table shows that our proposed CATOD is also compatible with different text-to-image encoders with notable performance. We will add these ablations to our final revisions.

---

> > ### Comment · Reviewer_PRf3 · 2024-08-12
> > **Post-rebuttal comments.**
> >
> > The authors have addressed my concerns.
> > I keep positive attitude towards this paper and increase my score by one.

---

> ### Author Response · Authors · 2024-08-05
>
> **Part 5: The impact of initial size/quality in CATOD (Q10)**
>
> Thanks for the advice! It is intriguing to ablate CATOD over the initial training set size/quality. In detail, we retest CATOD with different numbers of initial samples (10,20,50) and different quality (high-quality and random sampling), with 100 images sampled at last and 10 images selected per cycle. These experiments are tested with adaptor LoRA. The results are shown as follows:
>
> | Initial Setting         | Axolotl (CLIP) | Axolotl (CMMD) | Emperor Penguin Chick (CLIP) | Emperor Penguin Chick (CMMD) |
> | ----------------------- | -------------- | -------------- | ---------------------------- | ---------------------------- |
> | 10 HQ initial samples   | 73.99          | 0.84           | 74.97                        | 0.71                         |
> | 10 RAND initial samples | 72.89          | 0.93           | 73.95                        | 0.75                         |
> | 20 HQ initial samples   | 75.35          | 0.82           | 74.89                        | 0.69                         |
> | 20 RAND initial samples | 72.27          | 1.10           | 72.38                        | 0.73                         |
> | 50 HQ initial samples   | 75.37          | 0.81           | 74.56                        | 0.63                         |
> | 50 RAND initial samples | 70.39          | 1.34           | 68.85                        | 1.12                         |
>
> From these results, we may draw the following conclusions:
>
> 1. With HQ initial samples, the initial batch size does not have a significant compact, since a change in this size is not more than 0.3 in the CLIP score and 0.08 in the MMD score. However, this initial size has a significant impact with randomly initialized samples with at most a 5.71 loss in CLIP score and 0.49 in CMMD. We account this phenomenon for that randomly initialized images contain more bad-quality ones that mislead the adaptation.
> 2. The quality of initial samples does have an impact on generative results since we can see a consistent performance loss when changing HQ initial samples to randomly initialized samples. With the initial size increase, this impact tends to be even more significant.
>
> The full results will be added in our revisions.
>
> **Part 6: Evaluating the quality of ID concepts after adapting OOD concepts. (Q11)**
>
> In brief, our method will not cause much degradation in non-OOD concepts. To prove this, we list a comparison of the performance of the following non-OOD concepts (corresponding to Q1) over SD 2.0 when fine-tuning with LoRA. The results are shown in the table below:
>
> | Insect            | Thrips | Flea Beetle | Aphids | Red Spider | Meadow Moth |
> | ----------------- | ------ | ----------- | ------ | ---------- | ----------- |
> | **CLIP (before)** | 75.35  | 72.76       | 75.59  | 74.27      | 73.93       |
> | **CLIP (after)**  | 75.17  | 71.89       | 75.76  | 74.25      | 73.78       |
> | **CMMD (before)** | 2.05   | 1.28        | 1.16   | 1.85       | 1.47        |
> | **CMMD (after)**  | 2.08   | 1.32        | 1.30   | 1.93       | 1.49        |
>
> | Penguin           | Emperor Penguin | King Penguin | Little penguin | Magellanic Penguin | Adelie Penguin |
> | ----------------- | --------------- | ------------ | -------------- | ------------------ | -------------- |
> | **CLIP (before)** | 76.89           | 79.23        | 80.45          | 75.45              | 73.68          |
> | **CLIP (after)**  | 76.56           | 78.97        | 79.78          | 75.42              | 73.55          |
> | **CMMD (before)** | 1.19            | 0.75         | 1.46           | 1.60               | 0.77           |
> | **CMMD (after)**  | 1.26            | 0.80         | 1.58           | 1.61               | 0.86           |
>
> From this table, we can see that our proposed CATOD only leads to a performance loss of at most 0.15 on the CLIP score and at most 0.12 on CMMD on the non-OOD concepts after fine-tuning. This proves that our method does not distort non-OOD concepts when adapting native models to OOD concepts.

---

> ### Author Response · Authors · 2024-08-05
>
> **Part 7: Why CMMD is a good metric (Q12)**
>
> As reported in many recent works [1,2,3], most popular image matching metrics like Inception Score, Precision/Recall and FID (Frechet Inception Distance) may disagree with human raters, thus ill-suited for evaluating recent text-to-image models. In comparison, CMMD uses CLIP embeddings and Maximum Mean Discerpancy (MMD) that correctly ranks the image sets based on the severity of the distortion. The limitations of FID and previous metrics can be listed as follows:
>
> |                      | FID and other previous metrics                               | MMD distance                                                 |
> | -------------------- | ------------------------------------------------------------ | ------------------------------------------------------------ |
> | Inception embeddings | × Weak image embeddings<br />× Normality assumption<br />× Sample inefficient<br />× Biased estimator | × Weak image embeddings<br />√  Distribution-free<br />√  Sample inefficient<br />√  Unbiased estimator |
> | CLIP embeddings      | √ Rich image embeddings<br />× Normality assumption<br />× Sample inefficient<br />× Biased estimator | √  Rich image embeddings<br />√  Distribution-free<br />√  Sample inefficient<br />√  Unbiased estimator |
>
> [1] Jayasumana, Sadeep, Srikumar Ramalingam, Andreas Veit, Daniel Glasner, Ayan Chakrabarti, and Sanjiv Kumar. "Rethinking fid: Towards a better evaluation metric for image generation." *arXiv preprint arXiv:2401.09603* (2023).
>
> [2] Grimal, Paul, Hervé Le Borgne, Olivier Ferret, and Julien Tourille. "TIAM-A Metric for Evaluating Alignment in Text-to-Image Generation." In *Proceedings of the IEEE/CVF Winter Conference on Applications of Computer Vision*, pp. 2890-2899. 2024.
>
> [3] Lee, Tony, Michihiro Yasunaga, Chenlin Meng, Yifan Mai, Joon Sung Park, Agrim Gupta, Yunzhi Zhang et al. "Holistic evaluation of text-to-image models." *Advances in Neural Information Processing Systems* 36 (2024).
>
> **Part 8: Other minor questions**
>
> **Q1 (About Figure 1):** Yes, the CMMD in Figure 1 is calculated before adapting. This is used to show how out-of-distribution a concept is before adapting.
>
> **Q2 (About Figure 2):** Yes, the high-quality samples in the third part of Figure 2 are selected by the CATOD framework.
>
> **Q4 (About Eq. 6):** Apologies for the confusion! Since both the aesthetic and concept-matching scores range from 0 to 10, Equation 6 monotonically increases according to both variables, with both sine functions ranging from 0 to 1. We will ensure that the scores $\gamma_{aes}$ and $\gamma_{con}$ are clearly stated to range from 0 to 10 in future revisions.
>
> **Q6 (About the epochs to use):** Training for 20 epochs is common for adaptors such as LoRA, DreamBooth, and other related methods, as they typically require much less data (usually around 100 samples per concept). Training for too many epochs can lead to the underlying model replicating the images as exact copies.
>
> **Q7 (About algorithm 1):** We don't need to select new samples at each training epoch. The selection is performed in lines 2-6, prior to training. The algorithm demonstrates the process of a single Active Learning cycle, while training CATOD involves several cycles. The selection is conducted at the beginning of each cycle.
>
> **Q8 (About the schedule):** R includes 5 learning rates: $5\times 10^{-4},2.5\times  10^{-4},7.5\times10^{-5},5\times 10^{-5},2.5\times 10^{-5}$. Following Q7, we calculate the indicator $\gamma(A)$ after every epoch. When this indicator falls below the previous evaluation, we reduce the learning rate. If the learning rate cannot be lowered further, we conclude that the model has converged and the training is complete.
>
> **Part 9: Some typo errors  (According to Q3)**
>
> Thanks for pointing out the mistake! In lines 173 and 174, the sentence should be "if $r_k$ is the closet representation for $r_x$".

---

### Official Review · Reviewer_yLUC · 2024-07-12

**Soundness:** 3
**Presentation:** 2
**Contribution:** 3
**Rating:** 5
**Confidence:** 2

**Summary:**

This work presents CATOD, a new method for dealing with the challenging scenario of adapting generative models to OOD concepts. In particular, CATOD leverages an active learning setting to update the adaptor to generate better OOD images more broadly. Furthermore, the authors provide both extensive empirical evaluations and theoretical analysis which further support the validity of the CATOD methodology.

**Strengths:**

Strength:
- This paper provides a timely analysis of the difficult problem of adapting generative models to OOD concepts which is often overlooked in recent literature.
- CATOD showcases both strong empirical performance (CLIP score and CMMD score) on a reasonable set of 25 OOD concepts.
- Additionally, the paper provides strong theoretical links between the introduced aesthetic/concept-matching score and their necessity to the performance of adaptors.

**Weaknesses:**

The primary concern with this paper is the sparse empirical evaluations presented for CATOD. In particular, the 25 OOD concepts chosen seem few with only a small set of 100 samples left out for validation.

**Questions:**

The reviewer would like to have further clarification regarding the experimental setup of the paper. In particular, the reviewer would like to hear the author more directly address whether the small evaluation set used for the experiments is a reasonable concern.

**Limitations:**

The authors don't provide any full discussion of the limitations of the proposed methodology. Additionally, no direct comment has been made regarding societal impacts, however, the reviewer does not see any potential negative impacts derived from this work.

---

> ### Author Rebuttal · Authors · 2024-08-05
>
> Thanks for your comments!
>
> **Part 1: About the number of samples to use for training and validation.**
>
> In brief, since most recent adaptors require only a small number of training samples (usually around 100), our validation set is also set to be of the same scale as the training samples.
>
> As mentioned in the paper, *adaptors* are designed to reduce training costs when introducing new concepts. Specifically, they require significantly fewer samples compared to direct fine-tuning, typically no more than 1000 samples, and often around 100 samples. The following table provides a comparison of the most commonly used adaptors (including our proposed CATOD):
>
> | Aspect                       | Textual Inversion [1]                                     | DreamBooth [2]                                             | LoRA [3,4]                                                   | CATOD (Ours)                                             |
> | ---------------------------- | --------------------------------------------------------- | ---------------------------------------------------------- | ------------------------------------------------------------ | -------------------------------------------------------- |
> | **Training Data**            | Requires a few images (5-20)                              | Requires a large amount of labeled data (100-1000)         | Requires intermediate amount of data (around 100)            | Requires intermediate amount of data (around 100)        |
> | **Flexibility**              | High flexibility, but usually limited to only one concept | Not flexible, can learn multiple detailed attributes       | High flexibility, can learn multiple detailed attributes     | High flexibility, can learn multiple detailed attributes |
> | **Model Size**               | No change to model size                                   | Model size can increase                                    | Slight increase due to additional adaptation layers          | Depend on the adaptor, usually slight increalowse        |
> | **Quality for New Concepts** | Usually not good                                          | Good for ID concepts, but usually deteriorates on OOD ones. | Good for both ID concepts, but usually deteriorates on OOD ones. | Good for both ID and OOD concepts.                       |
>
> Since previous adaptors often fail to accurately depict out-of-distribution (OOD) concepts due to inconsistent image quality, we also ensure that our validation set is of high quality to provide accurate validation.
>
> [1] Gal, Rinon, Yuval Alaluf, Yuval Atzmon, Or Patashnik, Amit H. Bermano, Gal Chechik, and Daniel Cohen-Or. "An image is worth one word: Personalizing text-to-image generation using textual inversion." *arXiv preprint arXiv:2208.01618* (2022).
>
> [2] Ruiz, Nataniel, Yuanzhen Li, Varun Jampani, Yael Pritch, Michael Rubinstein, and Kfir Aberman. "Dreambooth: Fine-tuning text-to-image diffusion models for subject-driven generation." In *Proceedings of the IEEE/CVF conference on computer vision and pattern recognition*, pp. 22500-22510. 2023.
>
> [3] Hu, Edward J., Yelong Shen, Phillip Wallis, Zeyuan Allen-Zhu, Yuanzhi Li, Shean Wang, Lu Wang, and Weizhu Chen. "Lora: Low-rank adaptation of large language models." *arXiv preprint arXiv:2106.09685* (2021).
>
> [4] Yeh, Shih-Ying, Yu-Guan Hsieh, Zhidong Gao, Bernard BW Yang, Giyeong Oh, and Yanmin Gong. "Navigating text-to-image customization: From lycoris fine-tuning to model evaluation." In *The Twelfth International Conference on Learning Representations*. 2023.
>
>
>
> **Part 2: About the number of categories to use for the adaptors.**
>
> Thank you for reminding me to investigate the capacity of adaptors in learning categories with OOD concepts! Extending CATOD to encompass even more categories is straightforward. Using the category "Insect" as an example, we explored the impact of adding three additional OOD concepts. Specifically, we conducted experiments with the following additional categories: "Chlumetia transversa," "Mango flat beak leafhopper," and "Rhytidodera bowrinii white." The table below shows the average performance in the "Insect" category as we add new concepts:
>
> | Number of Categories            | DreamBooth + CATOD (CLIP $\uparrow$) | DreamBooth +for CATOD (CMMD$\downarrow$) | LoRA + CATOD (CLIP$\uparrow$) | LoRA + CATOD (CMMD$\downarrow$) |
> | ------------------------------- | ------------------------------------ | ---------------------------------------- | ----------------------------- | ------------------------------- |
> | 5                               | 70.83                                | 1.39                                     | 71.19                         | 1.13                            |
> | 6 (+Chlumetia transversa)       | 69.23                                | 1.57                                     | 68.05                         | 1.52                            |
> | 7 (+Mango flat beak leafhopper) | 66.35                                | 1.96                                     | 65.39                         | 2.03                            |
> | 8 (+Rhytidodera bowrinii white) | 64.07                                | 2.65                                     | 63.28                         | 2.77                            |
>
> From this table, we can see that while CATOD can be easily generalized to more concepts, it experiences a performance decline as the number of categories increases. This decline is because recently proposed adaptors were originally designed to accommodate only a few concepts. However, since CATOD is designed as a general framework, it can be easily adapted to work with any future adaptors that are capable of handling a larger number of concepts.

---

### Decision · Program_Chairs · 2024-09-25

**Decision:**

Accept (poster)

**Comment:**

The paper aims to improve the adaptation of large-scale text-to-image generative models that allows them to accurately depict the visual
details of out-of-distribution concepts. Inspired by the empirical evidence that the failure cases usually correlate with the low quality of training data, the paper proposes to leverage an active learning paradigm which couples high-quality data accumulation and
adaptor training, enabling a finer-grained enhancement of generative results. A novel weighted scoring system is developed to automatically balance between aesthetics and concept-matching, the two key factors that impact the quality of synthetic results. The overall contribution is supported by both theoretical analysis and empirical results.

Reviewers provided constructive feedback that helped the authors further improve the quality of the paper though the rebuttal process. At the end of the reviewer-author discussion period, all reviewers were in favor of accepting the paper. The authors are encouraged to incorporate the reviewers' feedback into the final version of the paper.